# Communication-efficient Random-Walk Optimizer for Decentralized Learning

## Abstract

Decentralized learning has gained popularity due to its flexibility and the ability to operate without a central server. A popular family of decentralized learning methods is based on random-walk optimization, which is easy to implement and has a low computation cost. However, random-walk optimization with adaptive optimizers can suffer from high communication cost. In this paper, we propose to address this problem from three directions. First, we eliminate the communication of auxiliary parameters, such as momentum and preconditioner, in adaptive optimizers. We also perform multiple model updates on the same client before sending the model to next client. Additionally, we extend sharpness-aware minimization (SAM) to random-walk optimization to avoid overfitting on local training data. Our theoretical analysis demonstrates that the proposed method can converge faster than existing approaches with the same communication cost. Empirical results on various datasets, communication networks, and network sizes show that the proposed method outperforms existing approaches while significantly reducing communication costs.

## 1 Introduction

With the explosive growth of data, it may not be feasible or desirable to store all the data at a single location due to storage limitations or privacy concerns. Distributed learning (Balcan et al., 2012; Yuan et al., 2022), which does not require gathering the data and allows training on large-scale data while maintaining data privacy, is thus highly desirable. It has found diverse applications in various fields such as Internet-of-Things (IoT) (Nguyen et al., 2021) and healthcare applications (Dayan et al., 2021; Ogier du Terrail et al., 2023).

Distributed learning methods can be classified as centralized methods (Predd et al., 2009; Balcan et al., 2012) and decentralized methods (Lian et al., 2017; 2018; Mao et al., 2020; Lu & De Sa, 2021; Sun et al., 2022). Centralized distributed learning still assumes the presence of a central server to coordinate the computation and communication during model training. However, this central server can become a bottleneck. The training process critically depends on the computation and communication capacity of the server, not to mention potentially catastrophic consequences when the server is broken. Moreover, it can be hard to find a central server that is trusted by all clients. Decentralized learning, on the other hand, does not involve a central server, and is thus more flexible and preferable. Decentralized learning methods can be further subdivided as gossip methods (Lian et al., 2017; 2018; Koloskova et al., 2020) and random-walk methods (Mao et al., 2020; Sun et al., 2022; Triastcyn et al., 2022). Gossip methods activate all clients in each round, while random-walk (or incremental) methods activate only one client in each round. While many works consider gossip methods, requiring all clients to be always active can be difficult in practice. For example, in IoT applications, clients can be offline due to energy or communication issues, especially when they are placed in the wild. Hence, in this paper, we focus on random-walk methods.

In machine learning, the learning process can often be expedited with the use of adaptive optimizers such as AdaGrad (Duchi et al., 2011) or Adam (Kingma & Ba, 2015). However, when used with random-walk methods, auxiliary parameters of these adaptive optimizers (e.g., momentum and preconditioner) also need to be communicated. This can lead to a huge increase in the communication cost, as these auxiliary parameters have the same size as the model parameter.

To alleviate this problem, we propose a simple but effective optimizer called Localized SAM (Sharpness-Aware Minimization) optimizer, which removes the communication of auxiliary parameters completely. To further reduce the communication cost, multiple model updates are performed on the active client before sending the model to the next client. This thus leads to a much smaller communication cost. We also incorporate sharpness-aware minimization (Foret et al., 2021) to random-walk optimization to avoid over-fitting on each client's local data with multiple updates. We prove theoretically that this modification can lead to faster convergence rate than existing methods. Empirical results demonstrate that the proposed method achieves better performances than various baselines and can significantly reduce the communication cost.

Our contributions can be summarized as follows:

- We propose a novel adaptive optimizer for decentralized random-walk optimization. By removing the communication of auxiliary parameters and performing multiple updates on the same client, the proposed optimizer achieves lower communication cost.

- We introduce sharpness-aware minimization to random-walk optimization, which alleviates potential over-fitting when performing multiple local updates on the same client.

- We show theoretically that the proposed method can converge faster than existing works under the same communication cost.

- We perform experiments on various data sets, communication networks and network sizes, demonstrating effectiveness of the proposed method.

## 2 RELATED WORKS

Given a set of $n$ clients, we consider solving the optimization problem (Predd et al., 2009; Balcan et al., 2012; Koloskova et al., 2020):

$$\min_{\boldsymbol{w}} \mathcal{L}(\boldsymbol{w}) = \sum_{i=1}^{n} \mathcal{L}_i(\boldsymbol{w}) = \sum_{i=1}^{n} \mathbb{E}_{\xi_i} \ell(\boldsymbol{w}, \xi_i) \tag{1}$$

in a decentralized manner. Here, $\boldsymbol{w}$ is the model parameter, $\xi_i$ is the training data on client $i$, and $\ell(\boldsymbol{w}, \xi_i)$ is client $i$'s loss on its local data. Existing decentralized learning algorithms can be classified as gossip algorithms (Lian et al., 2017; 2018; Xin et al., 2020; Koloskova et al., 2020; Lu & De Sa, 2021; Shi et al., 2023) or random-walk (incremental) algorithms (Mao et al., 2019; 2020; Sun et al., 2022; Triastcyn et al., 2022). In each round of a gossip algorithm, all clients need to be activated. Each client receives and aggregates updates from all its neighbors for model update. On the other hand, for random-walk algorithms, only one client is activated in each round. The active client receives current model from the previous client, and updates model parameters with its own training data, before sending updated model to the next client. The active client is selected from a Markov chain with transition probability matrix $\boldsymbol{P} = [\boldsymbol{P}_{ij}] \in \mathbb{R}^{n \times n}$, where $\boldsymbol{P}_{ij}$ is the probability $P(i_{t+1} = j \mid i_t = i)$ that the next client $i_{t+1}$ is $j$ given that the current client is $i$.

The pioneering work on random-walk decentralized optimization is (Bertsekas, 1997), which focuses only on the least squares problem. A more general algorithm in (Johansson et al., 2010) uses (sub-)gradient descent with Markov chain sampling. More recently, the Walkman algorithm (Mao et al., 2020) formulates problem (1) as a linearly-constrained optimization problem, which is then solved by the alternating direction method of multipliers (ADMM) (Boyd et al., 2011). However, these works do not consider adaptive learning rates or momentum in decentralized stochastic optimization.

In traditional machine learning, adaptive optimizers like AdaGrad (Duchi et al., 2011) or Adam (Kingma & Ba, 2015) have become popular alternatives to vanilla SGD. By adapting the learning rate based on historical gradient information, adaptive optimizers often yield faster convergence than vanilla SGD (Chen et al., 2019). Very recently, adaptive optimizers are also used in random-walk decentralized optimization (Sun et al., 2022; Triastcyn et al., 2022). For example, Sun et al (Sun et al., 2022) proposes the use of an Adam variant. However, its communication cost is three times that of SGD, as both the momentum and preconditioner (which are of the same size as the model parameter) need to be transmitted. To overcome this issue, Triastcyn et al. (2022) proposes to drop the momentum, and also compress the preconditioner before transmission.

# 3 COMMUNICATION-EFFICIENT RANDOM-WALK DECENTRALIZED LEARNING

In principle, any stochastic optimizer used in centralized learning can be directly applied to random-walk decentralized optimization (Sun et al., 2022). The main difference is that the sampling of centralized data is replaced by a random walk across different clients. An example based on the SGD optimizer is shown in Algorithm 1. Another example, based on the adaptive optimizer Adam (Kingma & Ba, 2015) is shown in Algorithm 2.[1] Algorithm 2 can often have faster convergence and better generalization than the SGD-based Algorithm 1, as will be demonstrated empirically in Section 4.1.

In each iteration of Algorithm 2, auxiliary parameters (momentum $m_t$ and preconditioner $v_t$) also need to be transmitted. Since each of them is of the same size as model parameter, the communication cost of Algorithm 2 is three times that of Algorithm 1. This can be critical when communication is not efficient (e.g., wireless communication in the wild). To alleviate this problem, Triastcyn et al. (2022) proposed to remove the momentum term $m_t$ from Adam, and compress the preconditioner $v_t$ before transmission. The communication cost is then reduced to $1 + c$ of that of SGD, where $c$ is the compression ratio (i.e., ratio of parameter sizes before and after compression) for the preconditioner $v_t$. In practice, the performance depends critically on the compression algorithm. Hence, $c$ cannot be too small, and so is the communication cost reduction.

---

**Algorithm 1** SGD-based random-walk decentralized learning.

1: **Input:** learning rate $\eta > 0$.
2: set the first client $i_0$
3: **for** $t = 0$ **to** $T - 1$ **do**
4:     compute $g_t = \nabla_{w_t} \ell(w_t; \xi_{i_t})$
5:     $w_{t+1} = w_t - \eta g_t$
6:     transmit model $w_{t+1}$ to next client $i_{t+1}$
7: **end for**

**Algorithm 2** Adam-based random-walk decentralized learning (Sun et al., 2022).

1: **Input:** learning rate $\eta > 0$, hyper-parameters $0 \leq \theta < 1, \delta > 0$.
2: set the first client $i_0$ and initialize $m_{-1} = \mathbf{0}, v_{-1} = \mathbf{0}$
3: **for** $t = 0$ **to** $T - 1$ **do**
4:     compute $g_t = \nabla_{w_t} \ell(w_t; \xi_{i_t})$
5:     $m_t = \theta m_{t-1} + (1 - \theta) g_t$
6:     $v_t = v_{t-1} + [g_t]^2$
7:     $w_{t+1} = w_t - \eta \frac{m_t}{(v_t + \delta \mathbf{1})^{1/2}}$
8:     transmit $(w_{t+1}, m_t, v_t)$ to next client $i_{t+1}$
9: **end for**

---

## 3.1 PROPOSED METHOD

To reduce the communication cost, we propose to remove the communication of additional auxiliary parameter entirely. As is theoretically analyzed in (Wang et al., 2022), removing the momentum in adaptive optimizers can still achieve comparable generalization performances with the original optimizer that uses momentum. Therefore, we propose to remove the momentum term in decentralized optimizer, which also removes the communication cost for sending current momentum to other clients. We then further let each client $i$ keeps its own preconditioner $v^i$. These local preconditioners are no longer sent to other clients, and are updated only when this specific client is activated. By further removing the communication of pre-conditioner, the communication cost is reduced to the same as that of Algorithm 1.

Despite that the communication of auxiliary parameter has been eliminated, the transmission of model parameters after each updates can still be expensive. To further reduce the communication cost, we perform multiple model updates on the same client before sending the model to the next client, as is similar to FedAvg (McMahan et al., 2017). The local preconditioner $v^{i_t}$ is also updated. Afterwards, the active client transmits only the model to the next client.

While the use of local updates can be useful for reducing the communication cost, it may overfit the training data as they are used by the same client for several times. To alleviate this problem, we propose to integrate sharpness-aware minimization (SAM) (Foret et al., 2021) to random-walk decentralized optimization. The key modification is that the gradient computed at weight $w_{Kt+k}$ is replaced by a perturbed model $w_{Kt+k} + \rho \frac{g_k'}{\|g_k'\|}$, where $g_k' = \nabla_{w_{Kt+k}} \ell(w_{Kt+k}; \xi_{i_t})$ is the gradient

---

[1]This is also called Adaptive Random Walk Gradient Descent in (Sun et al., 2022).

evaluated at $\boldsymbol{w}_{Kt+k}$. This perturbation avoids converging to a sharp minimizer with worse generalization performance. The complete algorithm, which will be called Localized SAM Optimizer, is shown in Algorithm 3.

---

**Algorithm 3** Localized SAM Optimizer.
1: **Input:** learning rate $\eta > 0$, hyper-parameters $\delta > 0, \rho \geq 0$ and $K$.
2: initialize $\boldsymbol{v}^i = \mathbf{0}$ for all client $i$
3: **for** $t = 0$ **to** $T - 1$ **do**
4:      initialize $\boldsymbol{v}_0^{i_t} = \boldsymbol{v}^{i_t}$
5:      **for** $k = 0$ **to** $K - 1$ **do**
6:          compute $\boldsymbol{g}_k' = \nabla_{\boldsymbol{w}_{Kt+k}} \ell(\boldsymbol{w}_{Kt+k}; \xi_{i_t})$
7:          compute $\boldsymbol{g}_k = \nabla_{\boldsymbol{w}_{Kt+k}} \ell\left(\boldsymbol{w}_{Kt+k} + \rho \frac{\boldsymbol{g}_k'}{\|\boldsymbol{g}_k'\|}; \xi_{i_t}\right)$
8:          $\boldsymbol{v}_{k+1}^{i_t} = \boldsymbol{v}_k^{i_t} + [\boldsymbol{g}_k]^2$
9:          $\boldsymbol{w}_{Kt+k+1} = \boldsymbol{w}_{Kt+k} - \eta \frac{\boldsymbol{g}_k}{(\boldsymbol{v}_{k+1}^{i_t} + \delta \mathbf{1})^{1/2}}$
10:      **end for**
11:      store $\boldsymbol{v}^{i_t} = \boldsymbol{v}_K^{i_t}$
12:      transmit $\boldsymbol{w}_{K(t+1)}$ to next client $i_{t+1}$
13: **end for**

---

### 3.2 COMMUNICATION COST

Table 1 compares the relative per-iteration communication costs (amount of transmitted data) of the following random-walk decentralized learning algorithms: (i) Algorithm 1, which is based on SGD; (ii) Algorithm 2 (Sun et al., 2022), which is based on Adam; (iii) Compressed Adam without momentum (Triastcyn et al., 2022); (iv) Walkman (Mao et al., 2020); and (v) the proposed Localized SAM optimizer (Algorithm 3). Algorithm 1 and Walkman only transmit the model parameter in each iteration, and their communication costs are taken as 1. Similar to (Triastcyn et al., 2022), Localized SAM also reduces the communication cost by $K$ times than directly using local momentum and preconditioner. Compared with (Triastcyn et al., 2022), Localized SAM still has a smaller communication cost as the compression ratio $c$ is nonzero.

Table 1: Relative per-iteration communication costs for various random-walk decentralized learning algorithms. For most experiments in (Triastcyn et al., 2022), $c$ is roughly 0.6.

| SGD/Walkman (Mao et al., 2020) | Adam (Sun et al., 2022) | Compressed Adam without momentum (Triastcyn et al., 2022) | Localized SAM |
|---|---|---|---|
| 1 | 3 | $(1+c)/K$ | $1/K$ |

### 3.3 CONVERGENCE ANALYSIS

In this section, we analyze the convergence properties of the proposed Localized SAM optimizer. We make the following assumptions on the noisy gradient and loss function, which are commonly used in theoretical analysis for decentralized learning algorithms (Koloskova et al., 2020; Lu & De Sa, 2021; Sun et al., 2022; Triastcyn et al., 2022).

**Assumption 3.1** (Lower-bounded loss). There exists a constant $\mathcal{L}^*$ such that $\mathcal{L}(\boldsymbol{w}) \geq \mathcal{L}^*$ for all $\boldsymbol{w} \in \mathbb{R}^d$.

**Assumption 3.2** (Bounded gradients). There exists a constant $G > 0$ such that $\|\nabla \ell(\boldsymbol{w}; \xi)\| \leq G$ for all $\boldsymbol{w} \in \mathbb{R}^d$.

**Assumption 3.3** (Lipschitz smoothness of loss). There exists a constant $M > 0$ such that $\|\nabla \ell(\boldsymbol{w}; \xi) - \nabla \ell(\boldsymbol{w}'; \xi)\| \leq M \|\boldsymbol{w} - \boldsymbol{w}'\|$ for all $\boldsymbol{w}, \boldsymbol{w}' \in \mathbb{R}^d$.

**Assumption 3.4** (Bounded gradient variance). There exists a constant $\sigma > 0$ such that $\mathbb{E}_\xi \|\nabla \ell(\boldsymbol{w}; \xi) - \nabla \mathcal{L}(\boldsymbol{w})\|^2 \leq \sigma^2$ for all $\boldsymbol{w} \in \mathbb{R}^d$.

**Assumption 3.5** (Bounded difference on client gradient). For any $\boldsymbol{w}$, there exists a constant $\varsigma > 0$ that upper bounds the distance between the gradient on any given client $i$ and the global gradient: $\|\nabla \mathcal{L}_i(\boldsymbol{w}) - \nabla \mathcal{L}(\boldsymbol{w})\| \leq \varsigma$.

**Assumption 3.6** (Bounded gradient variance on each client)**.** For any $\boldsymbol{w}$ and client $i$, there exists a constant $\sigma_l > 0$ such that for data $\xi_i$ sampled from the training data of client $i$, the gradient variance satisfies $\mathbb{E}_{\xi_i} \|\nabla\ell(\boldsymbol{w}, \xi_i) - \nabla\mathcal{L}_i(\boldsymbol{w})\|^2 \leq \sigma_l^2$.

From Assumptions 3.5 and 3.6, the gradient variance $\sigma^2$ is lower-bounded as: $\sigma^2 \geq \sigma_l^2 + \varsigma^2$.

**Theorem 3.7.** *Under Assumption 3.1 to 3.6, $\boldsymbol{w}_t$'s generated from Algorithm 3 satisfy that:*

$$\frac{1}{KT} \sum_{t=0}^{KT-1} \|\nabla\mathcal{L}(\boldsymbol{w}_t)\|^2$$
$$\leq \quad \mathcal{O}\left( \frac{\mathcal{L}(\boldsymbol{w}_0) - \mathcal{L}^*}{\eta KT} + \frac{\eta\sigma^2}{K} + \eta(1 - \frac{1}{K})(\sigma_l^2 + \varsigma^2 + G^2 + \rho^2) + \frac{\lambda\eta\sqrt{n}}{(1-\lambda)KT} \right).$$

*where we denote the eigenvalues $\{\lambda_i\}_{i=1}^n$ of transition matrix $\boldsymbol{P}$ as $1 = \lambda_1 > \lambda_2 > \cdots > \lambda_n > -1$ and $\lambda = \max(\lambda_2, |\lambda_n|)$.*

Proofs are in Appendix B. The above matches the convergence rate in (Triastcyn et al., 2022), despite that we have removed communication of the pre-conditioner. In other words, communication of the pre-conditioner does not affect the convergence rate. Regarding the dependency on the perturbation scale $\rho$ in SAM, Theorem 3.7 also matches existing theoretical analysis (Andriushchenko & Flammarion, 2022) on SAM in the centralized setting ($\mathcal{O}(\rho^2)$ on the convergence rate).

By decreasing $\eta$ with $K$ and $T$, the following Corollary shows that the bound goes to zero when $T \to \infty$.

**Corollary 3.8.** *On setting $\eta = \frac{1}{\sqrt{KT}}$. we have*

$$\frac{1}{KT} \sum_{t=0}^{KT-1} \|\nabla\mathcal{L}(\boldsymbol{w}_t)\|^2 \leq \mathcal{O}\left( \frac{\mathcal{L}(\boldsymbol{w}_0) - \mathcal{L}^*}{\sqrt{KT}} + \frac{\sigma^2}{\sqrt{K^3 T}} \right.$$
$$\left. + \frac{1}{\sqrt{KT}}(1 - \frac{1}{K})(\sigma_l^2 + \varsigma^2 + G^2 + \rho^2) + \frac{\lambda\sqrt{n}}{(1-\lambda)\sqrt{KT}} \right). \quad (2)$$

*When $K = 1$, we have*

$$\frac{1}{T} \sum_{t=0}^{T-1} \|\nabla\mathcal{L}(\boldsymbol{w}_t)\|^2 \leq \mathcal{O}\left( \frac{\mathcal{L}(\boldsymbol{w}_0) - \mathcal{L}^*}{\sqrt{T}} + \frac{\sigma^2}{\sqrt{T}} + \frac{\lambda\sqrt{n}}{(1-\lambda)\sqrt{T}} \right).$$

Recall that Algorithm 3 performs a total of $KT$ model updates and $T$ rounds of communication. As can be seen from (2), a larger $K$ allows faster convergence without changing the communication cost.

When setting $K = 1$, Algorithm 3 has the same $O(1/\sqrt{T})$ rate as centralized adaptive optimizers for non-convex objectives (Li & Orabona, 2019; Chen et al., 2019). This also matches the convergence rate of adaptive random-walk optimizer in Algorithm 2 of (Sun et al., 2022). However, Algorithm 3 is more advantageous in that its communication cost is much smaller (by removing the communication for both momentum and preconditioner).

## 4 EXPERIMENTS

In this section, we study the empirical performance of the proposed method on various data sets, communication networks and network sizes. The setup generally follows (Mao et al., 2020; Sun et al., 2022). We consider neural network training on CIFAR-10 and CIFAR-100 data sets. Following (Sun et al., 2022), we use ResNet-20 for CIFAR-10 and ResNet-32 for CIFAR-100. In both cases, training samples are uniformly divided among the clients. All experiments use the cross-entropy loss.

For the communication network, we use a ring graph and a 3-regular expander graph with 20 clients (Figure 1). In the ring graph, the clients form a ring; while in the 3-regular expander graph, all clients have 3 neighbors. These graphs have been popularly used in comparing both random-walk (Mao et al., 2020; Sun et al., 2022) and gossip-based algorithms (Zhu et al., 2022; Vogels et al., 2022).

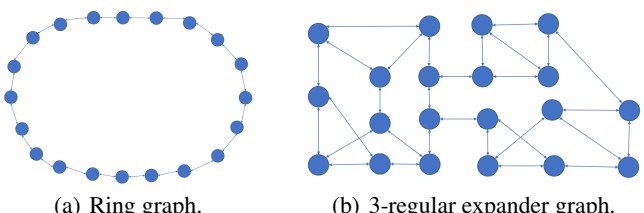

(a) Ring graph.      (b) 3-regular expander graph.

Figure 1: Communication networks used in the experiments.

The proposed Localized SAM Optimizer (Algorithm 3) is compared with random-walk decentralized optimization methods, including: (i) Algorithm 1, which is based on SGD, (ii) Algorithm 2 (Sun et al., 2022), which is based on Adam; (iii) Walkman (Mao et al., 2020); and (iv) the method in (Triastcyn et al., 2022), which is Adam-based without momentum (here, compression is not used, i.e., $\lambda = 1$ in Table 1). Recall that random-walk methods only need one active client in each round. For further comparison, we also consider baselines on (v) gossip optimization (Koloskova et al., 2020; Vogels et al., 2022) (with Adam as base optimizer), which activates all clients in each round, and (vi) federated averaging (FedAvg) (McMahan et al., 2017), which is a centralized distributed learning algorithm (i.e., requires a central server). In the experiment, we sample 4 clients in each FedAvg round. Each client updates its local model for 5 times before sending it back to the central server.

For SGD and Walkman (Mao et al., 2020), we set their learning rates to 0.05. For the other methods that are based on Adam, we set $\eta = 0.001$, $\theta = 0.9$ and $\delta = 0.00001$, which follows previous settings in Sun et al. (2022); Triastcyn et al. (2022). Unless otherwise stated, for Algorithm 3, the number of local updates $K$ is set to 5 and the inner stepsize $\rho$ is 0.05, and we have conducted sensitivity analysis on $K$ and $\rho$ in Section 4.3.

For performance evaluation, we report the training and testing losses with (i) computation cost, which is measured by the total number of weight updates, [2] and (ii) (relative) communication cost, which counts each sending of the model parameter as 1. The communication costs for random-walk methods are shown in Table 1. For the gossip method, since each client has to send its current model to all its neighbors in each iteration, the communication cost per iteration is equal to the number of edges in the communication network (i.e., $2n$ for the ring graph and $3n$ for the 3-regular graph, where $n$ is the number of clients). For FedAvg, in each round, each sampled client performs a number of local updates on its model and then sends the updated model to the server. The server then aggregates the local models (by computing the mean), and sends it back to the clients in the next round. Hence, the communication cost is two times (for downloading/uploading models) the number of clients used in each iteration divided by the number of local updates, which is $(2 \times 4)/5 = 8/5$ here.

### 4.1 Experiments with Different Communication Networks

Figure 2 presents a comparison of the training and testing losses, along with the number of model updates and communication cost, for various algorithms on CIFAR-10. In general, methods based on adaptive optimizers outperform SGD or Walkman, regardless of the type of communication (centralized, random-walk, or gossip). Moreover, the gossip method has a much higher communication cost than both centralized and random-walk methods. Walkman achieves slightly better performance than SGD with the same communication cost, but its performance is significantly worse than that of adaptive optimizers. Directly using Adam in random-walk optimization leads to higher communication cost than the other methods (Figures 2(c), 2(d), 2(g), 2(h)), and using Adam without momentum reduces communication cost while maintaining good performance. The proposed Localized SAM optimizer achieves similar convergence rate as other methods based on adaptive optimizers (Figures 2(a), 2(b), 2(e), 2(f)), while its communication cost is significantly smaller than all other methods (Figures 2(c), 2(d), 2(g), 2(h)).

Similarly, Figure 3 compares the training and testing losses on CIFAR-100 with the number of weight updates and communication cost for various algorithms. All methods based on adaptive optimizers have better performance than SGD or Walkman, and gossip method still has a much higher communication cost compared to both centralized and random-walk methods. Walkman achieves slightly better performance than SGD with the same communication cost, but its performance is

---

[2] For algorithm 3, after $k$ local updates on client $i_t$ at iteration $t$, the total number of weight updates is $Kt + k$.

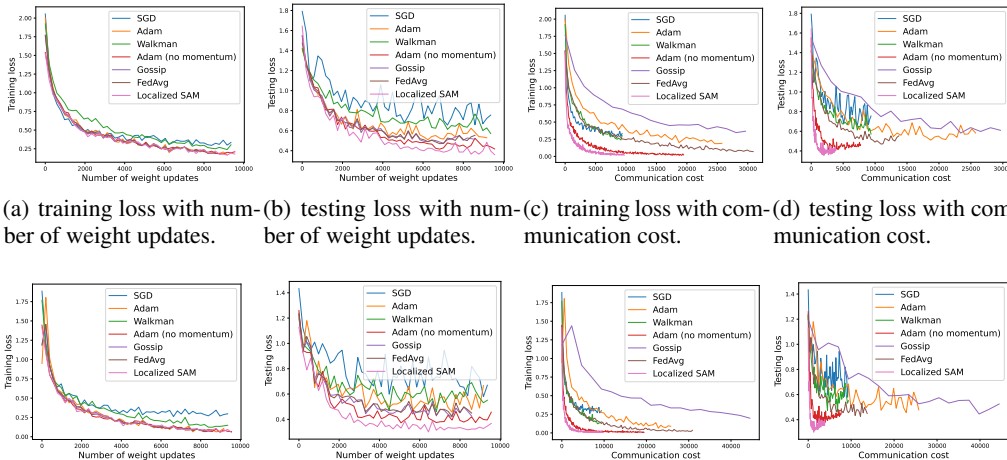

(a) training loss with number of weight updates.  (b) testing loss with number of weight updates.  (c) training loss with communication cost.  (d) testing loss with communication cost.

(e) training loss with number of weight updates.  (f) testing loss with number of weight updates.  (g) training loss with communication cost.  (h) testing loss with communication cost.

Figure 2: Training and testing losses for clients in a ring graph (top) and 3-regular expander graph (bottom), obtained by training ResNet-20 on CIFAR-10.

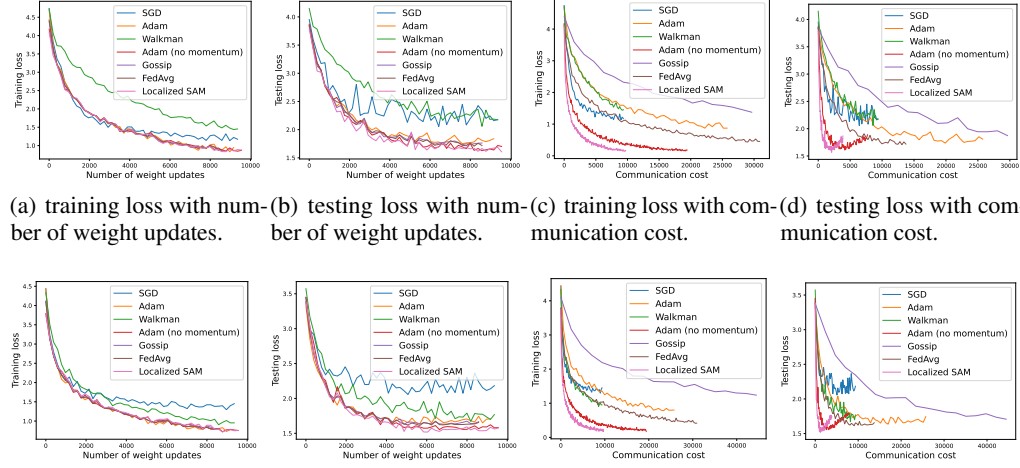

(a) training loss with number of weight updates.  (b) testing loss with number of weight updates.  (c) training loss with communication cost.  (d) testing loss with communication cost.

(e) training loss with number of weight updates.  (f) testing loss with number of weight updates.  (g) training loss with communication cost.  (h) testing loss with communication cost.

Figure 3: Training and testing losses for clients in a ring graph (top) and 3-regular expander graph (bottom), obtained by training ResNet-32 on CIFAR-100.

significantly worse than that of adaptive optimizers. Directly using Adam in random-walk optimization incurs a significantly higher communication cost compared to other methods (Figures 3(c), 3(d), 3(g), 3(h)). Using Adam without momentum reduces communication cost while maintaining good performances. The Localized SAM optimizer achieves the best overall performance among all methods (Figures 3(a), 3(b), 3(e), 3(f)), and its communication cost is significantly smaller than all other methods (Figures 3(c), 3(d), 3(g), 3(h)).

## 4.2 EXPERIMENTS WITH DIFFERENT NUMBERS OF CLIENTS

In this section, we study the impact of the number of clients on various random-walk decentralized optimization algorithms. We compare the training and testing losses on CIFAR-10 with the number of weight updates and communication cost for the ring graph (Figure 4) and 3-regular graph (Figure 5) with 10 or 50 clients. Similar to the results in Section 4.1 with 20 clients, all methods based on adaptive optimizers outperform SGD or Walkman. Among them, gossip method has a much larger communication cost that grows with the number of clients. Adam achieves faster convergence and better final results than SGD at the cost of larger communication cost, and using Adam without momentum can partially reduce the communication cost. Gossip algorithm achieves slightly better

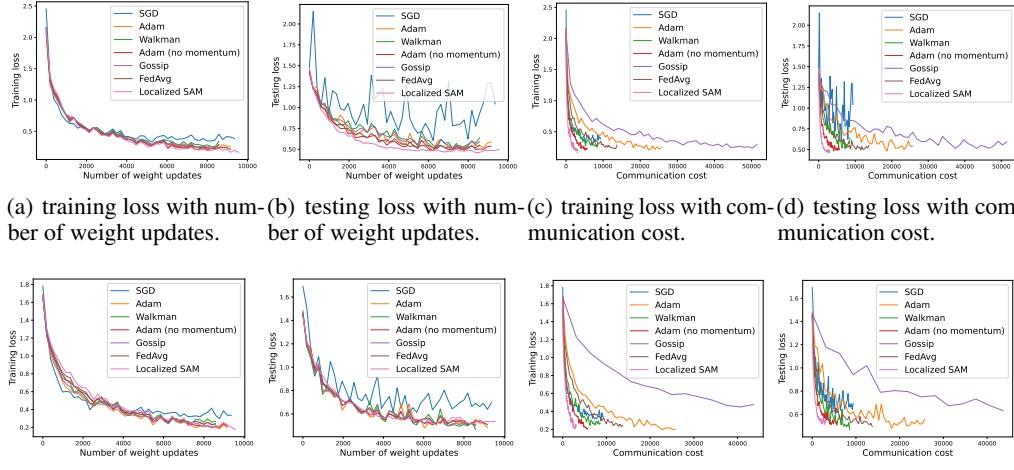

(a) training loss with num-ber of weight updates. (b) testing loss with num-ber of weight updates. (c) training loss with com-munication cost. (d) testing loss with com-munication cost.

(e) training loss with num-ber of weight updates. (f) testing loss with num-ber of weight updates. (g) training loss with communication cost. (h) testing loss with com-munication cost.

Figure 4: Training and testing losses for clients in a ring graph with 10 clients (top) and 50 clients (bottom), obtained by training ResNet-20 on CIFAR-10.

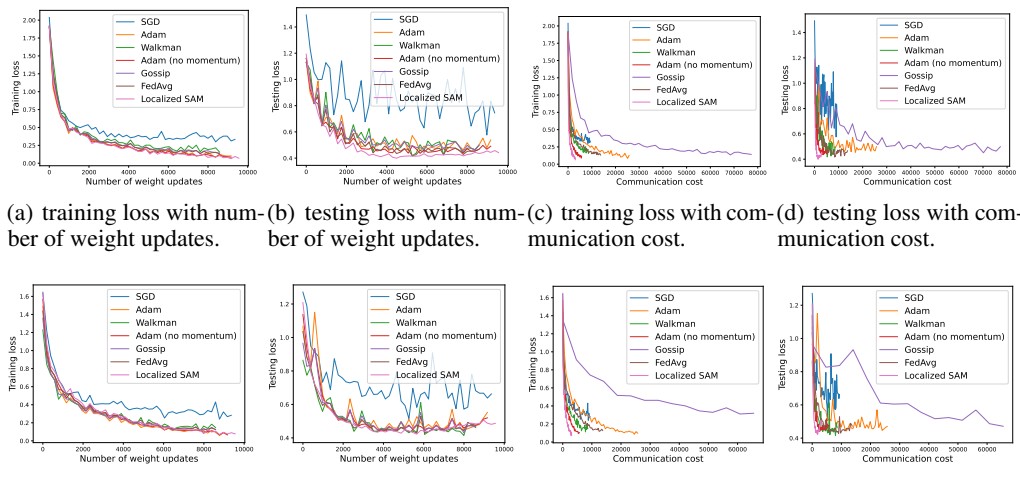

(a) training loss with num-ber of weight updates. (b) testing loss with num-ber of weight updates. (c) training loss with com-munication cost. (d) testing loss with com-munication cost.

(e) training loss with num-ber of weight updates. (f) testing loss with num-ber of weight updates. (g) training loss with communication cost. (h) testing loss with com-munication cost.

Figure 5: Training and testing losses for clients in a 3-regular graph with 10 clients (top) and 50 clients (bottom), obtained by training ResNet-20 on CIFAR-10.

performance than Adam, and its communication cost is smaller for the ring graph (Figures 4(c), 4(d), 4(g), 4(h)) and similar for the 3-regular graph (Figures 5(c), 5(d), 5(g), 5(h)). The proposed Localized SAM optimizer significantly reduces the communication cost compared to all other methods (Figures 4(c), 4(d), 4(g), 4(h) for ring graphs, Figures 5(c), 5(d), 5(g), 5(h) for 3-regular graphs) across different number of clients.

Figure 6 plots the training and testing losses obtained by Localized SAM during training with different numbers of clients. Similar to the other baselines (Mao et al., 2020; Sun et al., 2022), using more clients leads to slower convergence, which also agrees with intuition and the theoretical results in Theorem 3.7. Compared with the other methods in Figures 4 and 5, the proposed method still keeps the best performance among all baselines.

## 4.3 HYPERPARAMETER SENSITIVITY

In this experiment, we study the effect of the number of local updates $K$ and inner stepsize for SAM update $\rho$ on the performance of the localized SAM optimizer. We train a ResNet-20 on CIFAR-10 with 20-client on the ring and 3-regular expander graphs.

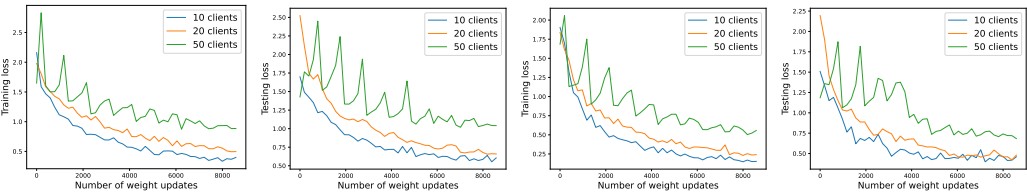

(a) Training loss on ring graph. (b) Testing loss on ring graph. (c) Training loss on 3-regular graph. (d) Testing loss on 3-regular graph.

Figure 6: Training and testing losses versus number of weight updates with different numbers of clients on CIFAR-10.

Figure 7 shows the training loss with $K \in \{1, 3, 5, 7, 9\}$. While a larger $K$ can lead to a larger communication cost reduction (Table 1), setting $K$ too large leads to worse models, which agrees with Theorem 3.7 and Corollary 3.8. Figures 7(c) and 7(d) demonstrate that setting $K = 5$ achieves a good trade-off between communication cost and convergence rate. Similar trends are obtained on the other data sets.

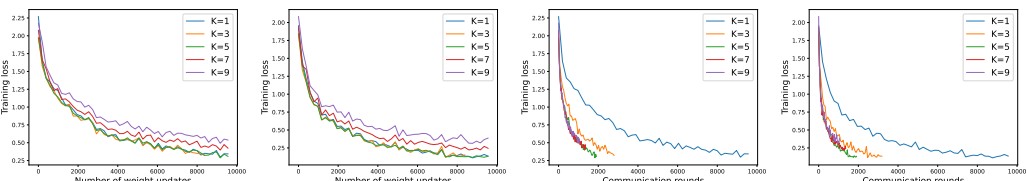

(a) loss with number of weight updates (ring graph). (b) loss with number of weight updates (3-regular graph). (c) loss with communication (ring graph). (d) loss with communication (3-regular graph).

Figure 7: Variation of training loss with number of weight updates (left) and communication cost (right) with different values of $K$.

Figure 8 shows the training and testing losses with $\rho \in \{0.0, 0.01, 0.02, 0.05, 0.1\}$. Note that $\rho$ does not affect the communication cost, and setting $\rho = 0.0$ reduces to the base optimizer (SGD or Adam). We can see that setting $\rho$ too small or too large lead to worse performance. A small $\rho$ (e.g., $\rho = 0.0$ or $\rho = 0.01$) cannot effectively escape sharp minimizers, while a large $\rho$ (e.g., $\rho = 0.1$) leads to unstable convergence. This agrees with the theoretical analysis (Andriushchenko & Flammarion, 2022) and experimental results on SAM in the centralized setting (Foret et al., 2021). In particular, setting $\rho$ to $0.05$ leads to the best overall performance, which matches with the recommended setting in the original SAM (Foret et al., 2021) (and is also the default used in previous experiments).

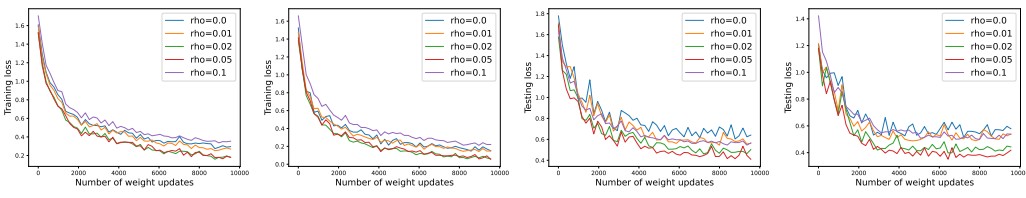

(a) Training loss on ring graph. (b) Training loss on 3-regular graph. (c) Testing loss on ring graph. (d) Testing loss on 3-regular graph.

Figure 8: Variation for training/testing loss with number of weight updates and different values of $\rho$.

## 5 CONCLUSION

In this paper, we propose a novel optimization algorithm, called Localized Adam, for random-walk decentralized learning. The proposed decentralized optimizer adopts adaptive learning rate, yet does not requires sending additional learning parameters to the next client in each iteration. Theoretically, we show that the proposed method still has the same convergence rate as existing stochastic optimizers. Experiments demonstrate that the proposed method achieves similar loss performance as various stochastic optimization baselines, but can significantly reduce the communication cost.

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

## A    ADDITIONAL EXPERIMENTAL RESULTS

### A.1    TESTING ACCURACY

In addition to the testing losses in the main text, Table 2 compares the testing accuracies obtained by different methods. These results are generally consistent with the loss performances: SGD and Walkman generally achieve slightly worse final performances than other methods. Among all methods that are based on adaptive optimizers, Localized SAM achieves the best overall performances.

Table 2: Final testing accuracy (in percents) for clients in different communication networks, obtained by training ResNet-20 on CIFAR-10.

| Graph type | Ring | | | 3-regular | | | Exponential | Chord |
|---|---|---|---|---|---|---|---|---|
| # clients | 10 | 20 | 50 | 10 | 20 | 50 | 20 | 20 |
| SGD | 82.3 | 81.7 | 81.4 | 83.2 | 82.5 | 81.9 | 82.8 | 82.7 |
| Adam | 85.2 | 85.0 | 84.6 | 88.8 | 88.1 | 87.4 | 88.3 | 88.0 |
| Walkman | 81.6 | 78.2 | 78.9 | 85.7 | 85.3 | 84.8 | 85.6 | 85.7 |
| Adam (no momentum) | 85.1 | 84.8 | 85.2 | 88.2 | 87.7 | 87.2 | 88.1 | 88.0 |
| Gossip | 85.6 | 85.3 | 84.9 | 88.5 | 88.2 | 87.8 | 88.5 | 88.4 |
| FedAvg | 85.4 | 85.5 | 85.0 | 88.3 | **88.6** | 87.6 | 88.5 | 88.3 |
| Localized SAM | **85.8** | **86.0** | **85.4** | **89.0** | **88.6** | **88.0** | **88.7** | **88.5** |

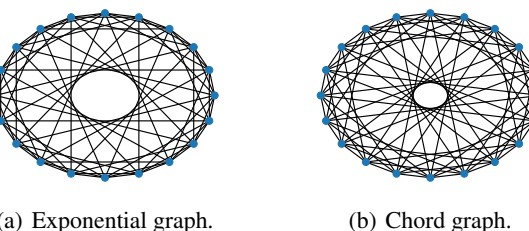

(a) Exponential graph.      (b) Chord graph.

Figure 9: Communication networks used in additional experiments.

## A.2 EXPERIMENTS WITH OTHER COMMUNICATION TOPOLOGIES

We have conducted experiments using exponential graphs Ying et al. (2021); Takezawa et al. (2023) and chord graphs Lian et al. (2018). The generation of exponential graph is the same as static exponential graph in Ying et al. (2021) with 20 clients, and the chord graph also follows Lian et al. (2018) with 20 clients.

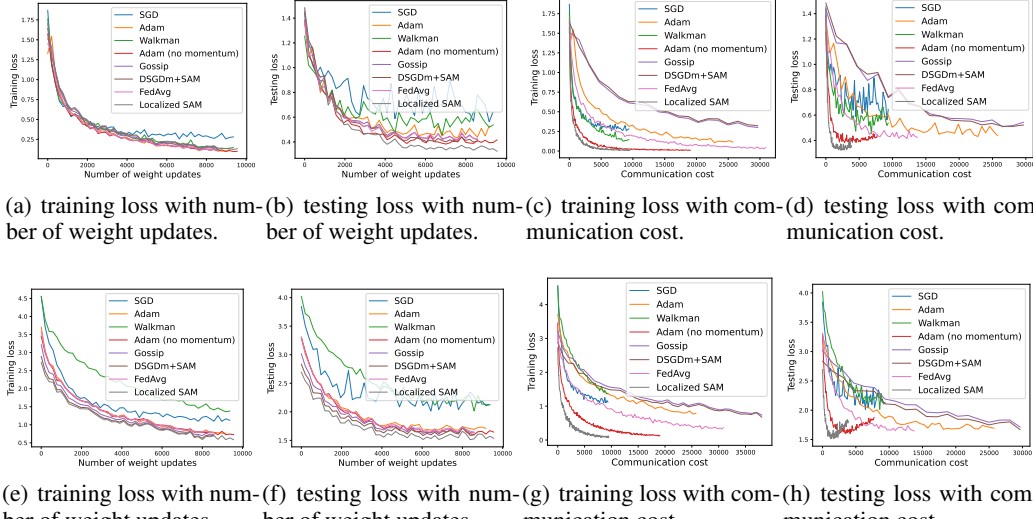

(a) training loss with number of weight updates.

(b) testing loss with number of weight updates.

(c) training loss with communication cost.

(d) testing loss with communication cost.

(e) training loss with number of weight updates.

(f) testing loss with number of weight updates.

(g) training loss with communication cost.

(h) testing loss with communication cost.

Figure 10: Training and testing losses for clients in an exponential graph with 20 clients, obtained by training ResNet-20 on CIFAR-10 (top)/CIFAR-100 (bottom).

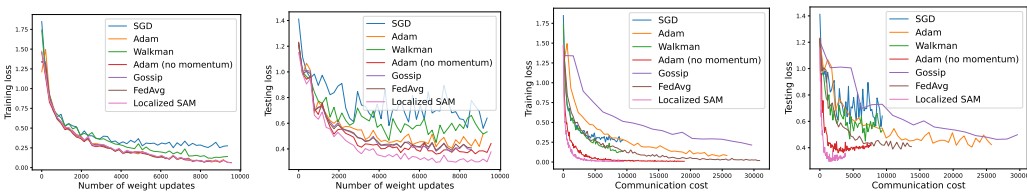

(a) training loss with num- (b) testing loss with num- (c) training loss with com- (d) testing loss with com-
ber of weight updates. ber of weight updates. munication cost. munication cost.

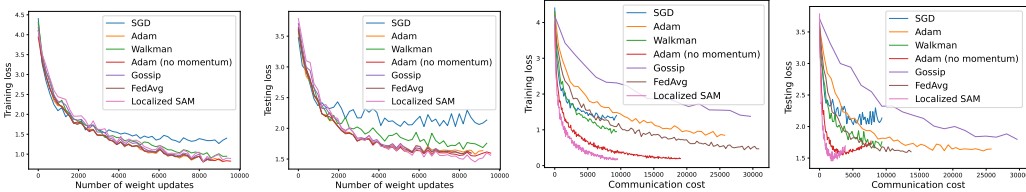

(e) training loss with num- (f) testing loss with num- (g) training loss with com- (h) testing loss with com-
ber of weight updates. ber of weight updates. munication cost. munication cost.

Figure 11: Training and testing losses for clients in a chord graph with 20 clients, obtained by training ResNet-20 on CIFAR-10 (top)/CIFAR-100 (bottom).

We have also considered another communication metric on exponential graph, which considers the average amount of communication for each client in each iteration, instead of the whole network. Under such metric, the (relative) communication costs for random-walk methods remain the same, since only one client communicate to other clients in each iteration. For the gossip method, the communication cost per iteration is now equal to the average number of edges (5 for exponential graph with 20 clients) for each client in the communication network. Results under this new metric, referred as communication cost (per client), are shown in Figure 12.

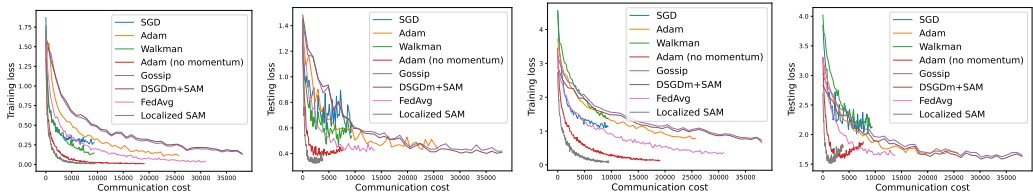

(a) CIFAR-10: training (b) CIFAR-10: testing loss (c) CIFAR-100: training (d) CIFAR-100: testing
loss with communication with communication cost. loss with communication loss with communication
cost. cost. cost.

Figure 12: Training and testing losses for clients in an exponential graph with 20 clients, obtained by training ResNet-20 on CIFAR-10/CIFAR-100.

## B    PROOFS

In this section, we prove Theorem 3.7 and Corollary 3.8 in Section 3.3. We first introduce the following lemma which will be useful in later proofs:

**Lemma B.1.** *Let $\boldsymbol{A} \in \{0,1\}^{n \times n}$ be an adjacency matrix of a connected, non-bipartite and undirected graph with $n$ nodes, and $\boldsymbol{P}$ be the transition matrix. Denote the marginal distribution of the random walk over the nodes at timestep $t$ as $\pi_t = \boldsymbol{P}^t \pi_0$, where $\pi_0$ is the initial distribution over all nodes. Then for any random variable $X$ that satisfies $|X| \leq G$, we have:*

$$|\mathbb{E}_{\pi_t}[X] - \mathbb{E}_{\pi^*}[X]| \leq G\sqrt{n}\lambda^t$$

*where we denote the eigenvalues $\{\lambda_i\}_{i=1}^n$ of transition matrix $\boldsymbol{P}$ as $1 = \lambda_1 > \lambda_2 > \cdots > \lambda_n > -1$, $\lambda = \max(\lambda_2, |\lambda_n|)$, $\pi^*$ is the stationary distribution.*

*Proof.* (Proof of Lemma B.1) We consider the total variation (TV) distances between $\pi_t$ and $\pi^*$. Denote the TV distance as $TV(\pi_t, \pi^*)$, and we can express it as:

$$TV(\pi_t, \pi^*) = \frac{1}{2}\|\pi_t - \pi^*\|_1 = \frac{1}{2}\|\boldsymbol{P}^t\pi_0 - \pi^*\|_1 = \frac{1}{2}\|\sum_{i=1}^{n} c_i\lambda_i^t\boldsymbol{b}_i - \pi^*\|_1$$

where we express the initial distribution $\pi_0 = c_i\boldsymbol{b}_i$ with the eigenvectors of matrix $\boldsymbol{P}$. Note that $\lambda_1 = 1$ and $\boldsymbol{b}_1 = \pi^*$, we have:

$$TV(\pi_t, \pi^*) = \frac{1}{2}\|\pi^* + \sum_{i=2}^{n} c_i\lambda_i^t\boldsymbol{b}_i - \pi^*\|_1 = \frac{1}{2}\|\sum_{i=2}^{n} c_i\lambda_i^t\boldsymbol{b}_i\|_1 \leq \frac{\sqrt{n}}{2}\|\sum_{i=2}^{n} c_i\lambda_i^t\boldsymbol{b}_i\|_2$$

Also, since $\boldsymbol{b}_i$'s are eigenvectors of $\boldsymbol{P}$, they should form an orthogonal basis. Thus, we have:

$$\|\sum_{i=2}^{n} c_i\lambda_i^t\boldsymbol{b}_i\|_2^2 = \sum_{i=2}^{n} c_i^2\lambda_i^{2t} \leq \lambda^{2t}\sum_{i=2}^{n} c_i^2 \leq \lambda^{2t}\sum_{i=1}^{n} c_i^2 = \lambda^{2t}$$

where $\lambda = \max(\lambda_2, |\lambda_n|)$. Then we have:

$$TV(\pi_t, \pi^*) \leq \frac{\sqrt{n}}{2}\lambda^t$$

Now by the definition of TV distance, we should have:

$$TV(\pi_t, \pi^*) = \frac{1}{2G}\sup_{|X|\leq G}(\mathbb{E}_{\pi_t}[X] - \mathbb{E}_{\pi^*}[X]) \leq \frac{\sqrt{n}}{2}\lambda^t$$

which gives us

$$|\mathbb{E}_{\pi_t}[X] - \mathbb{E}_{\pi^*}[X])| \leq G\sqrt{n}\lambda^t$$

and concludes our proof. $\square$

Now we are ready to present the proof for Theorem 3.7.

*Proof.* (Proof of Theorem 3.7). Similar to the case of $K = 1$, we have:

$$\mathcal{L}(\boldsymbol{w}_{t+1}) \leq \mathcal{L}(\boldsymbol{w}_t) + \nabla\mathcal{L}(\boldsymbol{w}_t)(\boldsymbol{w}_{t+1} - \boldsymbol{w}_t) + \frac{M}{2}\|\boldsymbol{w}_{t+1} - \boldsymbol{w}_t\|^2$$

$$= \mathcal{L}(\boldsymbol{w}_t) - \eta(\nabla\mathcal{L}(\boldsymbol{w}_t))^\top\frac{\boldsymbol{g}_t}{(\sqrt{\boldsymbol{v}_t} + \delta)} + \frac{\eta^2 M}{2}\|\frac{\boldsymbol{g}_t}{(\sqrt{\boldsymbol{v}_t} + \delta)}\|^2$$

Now we distinguish two cases: $t \mod K = 0$ or $t \mod K \neq 0$. When $t \mod K = 0$, the client is changed to another one by probability distribution $\pi_t$. From Assumption 3.4 and Lemma B.1, we have:

$$\mathbb{E}[\mathcal{L}(\boldsymbol{w}_{t+1})] - \mathcal{L}(\boldsymbol{w}_t) \leq -\eta(\nabla\mathcal{L}(\boldsymbol{w}_t))^\top\mathbb{E}\left[\frac{\boldsymbol{g}_t}{(\sqrt{\boldsymbol{v}_t} + \delta)}\right] + \frac{\eta^2 M}{2}\mathbb{E}\left\|\frac{\boldsymbol{g}_t}{(\sqrt{\boldsymbol{v}_t} + \delta)}\right\|^2$$

$$\leq -\eta\frac{\|\nabla\mathcal{L}(\boldsymbol{w}_t)\|^2}{(\sqrt{\boldsymbol{v}_t} + \delta)} + \frac{\eta(G + \eta M)}{\delta^2}\sigma^2 + \eta(\frac{\eta M}{2} + G)d\sqrt{n}\lambda^t$$

When $t \mod K \neq 0$, the client remains unchanged, so we have:

$$\mathbb{E}[\mathcal{L}(\boldsymbol{w}_{t+1})] - \mathcal{L}(\boldsymbol{w}_t) \leq -\eta(\nabla\mathcal{L}(\boldsymbol{w}_t))^\top\mathbb{E}\left[\frac{\boldsymbol{g}_t}{(\sqrt{\boldsymbol{v}_t} + \delta)}\right] + \frac{\eta^2 M}{2}\mathbb{E}\left\|\frac{\boldsymbol{g}_t}{(\sqrt{\boldsymbol{v}_t} + \delta)}\right\|^2 \quad (3)$$

Now we bound the two terms separately. For the first term in (3), we have:

$$-\eta(\nabla\mathcal{L}(\boldsymbol{w}_t))^\top\mathbb{E}\left[\frac{\boldsymbol{g}_t}{(\sqrt{\boldsymbol{v}_t}+\delta)}\right] = -\eta(\nabla\mathcal{L}(\boldsymbol{w}_t))^\top\mathbb{E}\left[\frac{\boldsymbol{g}_t}{(\sqrt{\boldsymbol{v}_t}+\delta)} + \frac{\boldsymbol{g}_t}{(\sqrt{\boldsymbol{v}_{t-1}}+\delta)} - \frac{\boldsymbol{g}_t}{(\sqrt{\boldsymbol{v}_{t-1}}+\delta)}\right.$$

$$\left. + \frac{\nabla\mathcal{L}(\boldsymbol{w}_t)}{(\sqrt{\boldsymbol{v}_{t-1}}+\delta)} - \frac{\nabla\mathcal{L}(\boldsymbol{w}_t)}{(\sqrt{\boldsymbol{v}_{t-1}}+\delta)}\right]$$

$$= -\eta\left\|\frac{(\nabla\mathcal{L}(\boldsymbol{w}_t))^2}{(\sqrt{\boldsymbol{v}_{t-1}}+\delta)}\right\|_1 - \eta(\nabla\mathcal{L}(\boldsymbol{w}_t))^\top\mathbb{E}\left[\frac{\boldsymbol{g}_t}{(\sqrt{\boldsymbol{v}_t}+\delta)} - \frac{\boldsymbol{g}_t}{(\sqrt{\boldsymbol{v}_{t-1}}+\delta)}\right]$$

$$- \eta(\nabla\mathcal{L}(\boldsymbol{w}_t))^\top\mathbb{E}\left[\frac{\boldsymbol{g}_t}{(\sqrt{\boldsymbol{v}_{t-1}}+\delta)} - \frac{\nabla\mathcal{L}(\boldsymbol{w}_t)}{(\sqrt{\boldsymbol{v}_{t-1}}+\delta)}\right]$$

$$\leq -\eta\left\|\frac{(\nabla\mathcal{L}(\boldsymbol{w}_t))^2}{(\sqrt{\boldsymbol{v}_{t-1}}+\delta)}\right\|_1 + \eta|\nabla\mathcal{L}(\boldsymbol{w}_t)|^\top\left|\mathbb{E}\left[\frac{\boldsymbol{g}_t}{(\sqrt{\boldsymbol{v}_t}+\delta)} - \frac{\boldsymbol{g}_t}{(\sqrt{\boldsymbol{v}_{t-1}}+\delta)}\right]\right|$$

$$+ \eta\left|\frac{(\nabla\mathcal{L}(\boldsymbol{w}_t))}{(\sqrt{\boldsymbol{v}_{t-1}}+\delta)}\right|^\top|\mathbb{E}[\boldsymbol{g}_t] - \nabla\mathcal{L}(\boldsymbol{w}_t)|$$

From the update rule of $\boldsymbol{v}_t = \boldsymbol{v}_{t-1} + \boldsymbol{g}_t^2$, we should have:

$$\frac{\boldsymbol{g}_t}{(\sqrt{\boldsymbol{v}_t}+\delta)} - \frac{\boldsymbol{g}_t}{(\sqrt{\boldsymbol{v}_{t-1}}+\delta)} \leq \frac{\boldsymbol{g}_t^2}{\delta(\sqrt{\boldsymbol{v}_{t-1}}+\delta)}$$

which gives us:

$$-\eta(\nabla\mathcal{L}(\boldsymbol{w}_t))^\top\mathbb{E}\left[\frac{\boldsymbol{g}_t}{(\sqrt{\boldsymbol{v}_t}+\delta)}\right] \leq -\eta\left\|\frac{(\nabla\mathcal{L}(\boldsymbol{w}_t))^2}{(\sqrt{\boldsymbol{v}_{t-1}}+\delta)}\right\|_1 + \frac{\eta G}{\delta}\left\|\mathbb{E}\left[\frac{\boldsymbol{g}_t^2}{\sqrt{\boldsymbol{v}_{t-1}}+\delta}\right]\right\|_1$$

$$+ \frac{\eta G}{\delta}\|\mathbb{E}[\boldsymbol{g}_t] - \nabla\mathcal{L}(\boldsymbol{w}_t)\|_1$$

$$\leq -\eta\left\|\frac{(\nabla\mathcal{L}(\boldsymbol{w}_t))^2}{(\sqrt{\boldsymbol{v}_{t-1}}+\delta)}\right\|_1 + \frac{\eta G}{\delta}\left\|\mathbb{E}\left[\frac{\boldsymbol{g}_t^2}{\sqrt{\boldsymbol{v}_{t-1}}+\delta}\right]\right\|_1$$

$$+ \frac{\eta G\sqrt{d}}{\delta}\|\mathbb{E}[\boldsymbol{g}_t] - \nabla\mathcal{L}(\boldsymbol{w}_t)\|_2$$

$$\leq -\eta\left\|\frac{(\nabla\mathcal{L}(\boldsymbol{w}_t))^2}{(\sqrt{\boldsymbol{v}_{t-1}}+\delta)}\right\|_1 + \frac{\eta G}{\delta}\left\|\mathbb{E}\left[\frac{\boldsymbol{g}_t^2}{\sqrt{\boldsymbol{v}_{t-1}}+\delta}\right]\right\|_1$$

$$+ \frac{\eta\sqrt{d}}{2\delta}(G^2 + \|\mathbb{E}[\boldsymbol{g}_t] - \nabla\mathcal{L}(\boldsymbol{w}_t)\|_2^2)$$

For the second term in (3), we have:

$$\frac{\eta^2 M}{2}\mathbb{E}\left\|\frac{\boldsymbol{g}_t}{(\sqrt{\boldsymbol{v}_t}+\delta)}\right\|^2 \leq \frac{\eta^2 M}{\delta}\left\|\mathbb{E}\left[\frac{\boldsymbol{g}_t^2}{(\sqrt{\boldsymbol{v}_t}+\delta)}\right]\right\|_1$$

Combine these two terms will then give us:

$$\mathbb{E}[\mathcal{L}(\boldsymbol{w}_{t+1})] - \mathcal{L}(\boldsymbol{w}_t) \leq -\eta\left\|\frac{(\nabla\mathcal{L}(\boldsymbol{w}_t))^2}{(\sqrt{\boldsymbol{v}_{t-1}}+\delta)}\right\|_1 + \frac{\eta G + \eta^2 M}{\delta}\left\|\mathbb{E}\left[\frac{\boldsymbol{g}_t^2}{\sqrt{\boldsymbol{v}_{t-1}}+\delta}\right]\right\|_1$$

$$+ \frac{\eta\sqrt{d}}{2\delta}(G^2 + \|\mathbb{E}[\boldsymbol{g}_t] - \nabla\mathcal{L}(\boldsymbol{w}_t)\|_2^2)$$

Now we need to bound $\left\|\mathbb{E}\left[\frac{\boldsymbol{g}_t^2}{\sqrt{\boldsymbol{v}_{t-1}}+\delta}\right]\right\|_1$, for which we have:

$$\left\|\mathbb{E}\left[\frac{\boldsymbol{g}_t^2}{\sqrt{\boldsymbol{v}_{t-1}}+\delta}\right]\right\|_1 \leq \frac{\|\mathbb{E}[\boldsymbol{g}_t]\|^2 + \sigma_t^2}{\sqrt{\boldsymbol{v}_{t-1}}+\delta} = \frac{\|\mathbb{E}[\boldsymbol{g}_t] - \nabla\mathcal{L}(\boldsymbol{w}_t) + \nabla\mathcal{L}(\boldsymbol{w}_t)\|^2 + \sigma^2}{\sqrt{\boldsymbol{v}_{t-1}}+\delta}$$

$$\leq \frac{2}{\sqrt{\boldsymbol{v}_{t-1}}+\delta}(\|\mathbb{E}[\boldsymbol{g}_t] - \nabla\mathcal{L}(\boldsymbol{w}_t)\|^2 + \|\nabla\mathcal{L}(\boldsymbol{w}_t)\|^2 + \frac{\sigma_t^2}{2})$$

Therefore we have:

$$
\begin{aligned}
\mathbb{E}[\mathcal{L}(\boldsymbol{w}_{t+1})] - \mathcal{L}(\boldsymbol{w}_t) \leq & -\eta \left\| \frac{(\nabla\mathcal{L}(\boldsymbol{w}_t))^2}{(\sqrt{\boldsymbol{v}_{t-1}}+\delta)} \right\|_1 + \frac{2\eta(G+\eta M)}{\delta}\left\| \frac{(\nabla\mathcal{L}(\boldsymbol{w}_t))^2}{\sqrt{\boldsymbol{v}_{t-1}}+\delta} \right\|_1 \\
& + \frac{2\eta(G+\eta M)}{\delta^2}(\|\mathbb{E}[\boldsymbol{g}_t]-\nabla\mathcal{L}(\boldsymbol{w}_t)\|^2 + \frac{\sigma_l^2}{2}) + \frac{\eta\sqrt{d}}{2\delta}(G^2+\|\mathbb{E}[\boldsymbol{g}_t]-\nabla\mathcal{L}(\boldsymbol{w}_t)\|_2^2) \\
\leq & -\frac{\eta}{4(G+\delta)}\|\nabla\mathcal{L}(\boldsymbol{w}_t)\|^2 \\
& + \frac{2\eta(G+\eta M)}{\delta^2}(\|\mathbb{E}[\boldsymbol{g}_t]-\nabla\mathcal{L}(\boldsymbol{w}_t)\|^2 + \frac{\sigma_l^2}{2}) + \frac{\eta\sqrt{d}}{2\delta}(G^2+\|\mathbb{E}[\boldsymbol{g}_t]-\nabla\mathcal{L}(\boldsymbol{w}_t)\|_2^2)
\end{aligned}
$$

Now summing from $t = 0$ to $KT - 1$ will give us:

$$
\begin{aligned}
\mathbb{E}[\mathcal{L}(\boldsymbol{w}_{KT})] - \mathcal{L}(\boldsymbol{w}_0) \leq & -\frac{\eta}{4(G+\delta)}\sum_{t=0}^{KT-1}\|\nabla\mathcal{L}(\boldsymbol{w}_t)\|^2 + +\frac{\eta(G+\eta M)}{\delta^2}\sigma^2 T + \eta(\frac{\eta M}{2}+G)\frac{\lambda d\sqrt{n}}{1-\lambda} \\
& + \frac{2\eta T(K-1)(G+\eta M)}{\delta^2}(\|\mathbb{E}[\boldsymbol{g}_t]-\nabla\mathcal{L}(\boldsymbol{w}_t)\|^2 + \frac{\sigma_l^2}{2}) \\
& + \frac{\eta\sqrt{d}T(K-1)}{2\delta}(G^2+\|\mathbb{E}[\boldsymbol{g}_t]-\nabla\mathcal{L}(\boldsymbol{w}_t)\|_2^2)
\end{aligned}
$$

Re-arranging these terms and multiplying both side by $\frac{4(G+\delta)}{\eta KT}$ gives us:

$$
\begin{aligned}
\frac{1}{KT}\sum_{t=0}^{KT-1}\|\nabla\mathcal{L}(\boldsymbol{w}_t)\|^2 \leq & \frac{4(G+\delta)(\mathcal{L}(\boldsymbol{w}_0)-\mathcal{L}^*)}{\eta KT} + \frac{4(G+\delta)(G+\eta M)\sigma^2}{\delta^2 K} + (\frac{\eta M}{2}+G)\frac{4(G+\delta)\lambda d\sqrt{n}}{(1-\lambda)KT} \\
& + \frac{8(K-1)(G+\eta M)(G+\delta)}{\delta^2 K}(\|\mathbb{E}[\boldsymbol{g}_t]-\nabla\mathcal{L}(\boldsymbol{w}_t)\|^2 + \frac{\sigma_l^2}{2}) \\
& + \frac{2\sqrt{d}(K-1)(G+\delta)}{\delta K}(G^2+\|\mathbb{E}[\boldsymbol{g}_t]-\nabla\mathcal{L}(\boldsymbol{w}_t)\|_2^2)
\end{aligned}
$$

Using the big-O notation, the above is equivalent to:

$$
\frac{1}{KT}\sum_{t=0}^{KT-1}\|\nabla\mathcal{L}(\boldsymbol{w}_t)\|^2 \leq \mathcal{O}(\frac{\mathcal{L}(\boldsymbol{w}_0)-\mathcal{L}^*}{\eta KT} + \frac{\eta\sigma^2}{K} + \eta(1-\frac{1}{K})(\sigma_l^2 + \varsigma^2 + G^2 + \rho^2) + \frac{\lambda\eta\sqrt{n}}{(1-\lambda)KT})
$$

which concludes our proof. $\square$

## C  FUTURE DIRECTIONS

A possible future direction will be adapting our proposed method to dynamic network topology. Such adaptation should be straight-forward as we only need the local neighborhood information in each iteration. Regarding convergence analysis under dynamic network topology, currently we assume the existence of a stationary distribution $\pi^*$, and the distribution $\pi_t$ on all clients at time $t$ has bounded TV distance from $\pi^*$. Such condition may need to be generalized to dynamic network topology by imposing certain assumptions on how the communication network changes with time.

Another direction is to combine the proposed method with model parameter compression, which can further reduce the communication cost. While such combination is also straight-forward, convergence analysis may also need to bound the difference for models before and after compression, and we should still attain the same asymptotic convergence rate with bounded compression error.

