# OpenReview forum: "Communication-efficient Random-Walk Optimizer for Decentralized Learning"
_ICLR.cc/2024/Conference — Submitted to ICLR 2024_

### Official Review · Reviewer_na6d · 2023-10-30

**Soundness:** 3 good
**Presentation:** 3 good
**Contribution:** 2 fair
**Rating:** 6
**Confidence:** 3

**Summary:**

The paper "Communication-efficient Random-Walk Optimizer for Decentralized Learning" proposes a variant of ADAM to solve decentralized problems in which communication exchange happens via random walk agent selection. The main contribution is to propose a variant of ADAM that only requires the exchange of the iterate itself as opposed to requiring the exchange of ADAM's momentum terms. This effectively divides the communication cost by a factor of three. The authors establish convergence results and provide numerical simulations to illustrate the performance of the proposed scheme.

**Strengths:**

The proposed work reduces communication cost of the combination of ADAM and a random walk procedure for information exchange by a factor 3 by omitting the transmission of ADAM's momentum terms. Additionally, to further save communications the authors have each agent run a mini-batch of tunable length K to further save communications. To avoid that this leads to overfitting the authors add a slight perturbation when calling the gradient oracle. The introduction of these modifications allows the authors to claim that the incurred sub-optimality, measured via the gradient magnitude, as they are dealing with non-convex problems, vanishes at the appropriate rate.

**Weaknesses:**

The work combines existing understood tools to provide with a new scheme. While many times such a combination is non-obvious, I would suggest the authors point out in the main text  the challenges faced in the analysis to obtain the final result.

**Questions:**

- Given that a mini-batch is introduced to reduce the communication cost by a factor of K, why is it necessarily the case that using this variant of ADAM which costs "1" to communicate  as opposed to "3" (original ADAM-based random-walk), better than using ADAM + random walk with a mini-batch size of 3K?
- From the simulations, it seems that ADAM performs worse on the training data set (counting only computations). Can the authors explain why this seems to be the case?
- I would suggest the authors comment on the scaling of the main result with network related parameters, and how these compare to the literature, i.e. local gradient variance, gradient heterogeneity, number of agents, etc.

---

> ### Author Response · Authors · 2023-11-20
> **Responses**
>
> We would like to thank you for your constructive feedback. Here we answer your questions point by point:
>
> > The work combines existing understood tools to provide with a new scheme. While many times such a combination
> is non-obvious, I would suggest the authors point out in the main text the challenges faced in the analysis to obtain
> the final result.
>
> Thank you for your suggestion. While our method includes some existing understood tools, we would
> like to clarify that a key component of our method is to use localized momentum and pre-conditioner, which has not been
> explored in previous works (Sun et al. 2022; Triastcyn et al. 2022) on random-walk decentralized optimization. Please also
> see our general responses regarding novelty. Using localized momentum and pre-conditioner creates additional difficulty in
> theoretical analysis, as the momentum and pre-conditioner are not always updated and synchronized.
>
> > Given that a mini-batch is introduced to reduce the communication cost by a factor of K, why is it necessarily the
> case that using this variant of ADAM which costs ”1” to communicate as opposed to ”3” (original ADAM-based
> random-walk), better than using ADAM + random walk with a mini-batch size of 3K?
>
> We suppose $K$ refers to the
> number of local steps in the inner loop of Algortihm 3, whose influence is studied in Figure 7, Section 4.3. While using a
> larger $K$ can reduce the communication cost, setting it too large can lead to worse models, which agrees with Theorem 3.7
> and Corollary 3.8. Therefore, solely using more local steps may not always lead to better performance, while removing the
> communication of auxiliary parameters (momentum or preconditioner) can be helpful in such case.
>
> > From the simulations, it seems that ADAM performs worse on the training data set (counting only computations).
> Can the authors explain why this seems to be the case?
>
> From Figure 2-5, it seems that Adam has almost the same
> performance as Gossip, FedAvg or Localized SAM (comparing the orange line with purple, gray and pink lines).
>
> > I would suggest the authors comment on the scaling of the main result with network related parameters, and how these compare to the literature, i.e. local gradient variance, gradient heterogeneity, number of agents, etc.
>
> In theorem 3.7, the convergence rate is related to the communication network with $\lambda$, which depends on the communication network
> (namely the eigenvalues of the transition matrix).
> The local gradient variance, denoted as $\sigma^2$, is assumed in Assumption 3.4. The gradient heterogeneity, as measured by differences on client gradients $\varsigma$,
> is assumed in Assumption 3.5.
> Both $\sigma$ and $\varsigma$ appear in theorem 3.7 as well
> and the dependency matches previous analysis in literature
> (Sun et al. 2022; Triastcyn et al. 2022).
> The number of agents (denoted as $n$) also appears in theorem 3.7,
> and its squared root dependency to convergence rate also matches previous analysis on random-walk decentralized algorithms (Triastcyn et al. 2022).

---

### Official Review · Reviewer_w8Ss · 2023-11-02

**Soundness:** 2 fair
**Presentation:** 2 fair
**Contribution:** 2 fair
**Rating:** 3
**Confidence:** 5

**Summary:**

This paper proposes a random-walk Adam-based optimizer for decentralized learning in the setting that only one agent is active at each communication round. The objective of the proposed algorithm is to reduce communication cost while still maintaining acceptable learning performance, which is achieved by removing the auxiliary momentum parameter, avoiding transmitting the preconditioner, and performing multiple local updates. To overcome the potential overfitting issue brought by multiple local updates, sharpness-aware minimization is adopted. Empirically and theoretical analysis are conducted.

**Strengths:**

The problem identified is of interest. The paper is in general well written and easy to follow. The proposed algorithm does save communication cost compared with the vanilla random walk Adam algorithm.

**Weaknesses:**

This paper mostly combines standard algorithms (random walk, Adam without momentum, SAM), although this is not a problem, the theoretically analysis needs to be improved. Meanwhile, the experimental part lacks new insights except for some expected results.

Major comments:

1.	Theorem 3.7 looks like a direct combination of theoretical results obtained by Adam without momentum and SAM. Furthermore, the proof in Appendix does not consider the convergence guarantee that could be achieved by random walk method. That is, the Markov chain is not considered. Note that the last equation in Page 13 is almost the same as the convergence result (Theorem 4.3, Triastcyn et al., 2022)) except it does not have the compression part. The proof also follows exactly as Triastcyn et al. (2022). The perturbed model is not used, means that sharp awareness minimization is not analyzed, which makes me question the soundness of Theorem 3.7.

2.	Since SAM is integrated to prevent potential overfitting, the experiment should present this effect compared with its counterpart that does not have the perturbed model. The lack of this experiment comparison would question the necessity of incorporating SAM in the proposed Localized framework.

3.	The simulation only shows the loss performance of the proposed algorithms and the benchmarks, however, in practical, we would be more interested to see the classification accuracy.

4.	The proposed algorithm is compared with FedAvg, however, for FedAvg, not all agents are communicating all the time, which does not make sense in the setting that FedAvg does not need to consider communication. That means, I suppose that if all agents in FedAvg communicate all the time, the performance of FedAvg might be much better than all the other methods, since there exists a coordinator, although the communication cost would be very high. The figures presented, however, show that Localized SAM is always better than FedAvg in the random sample setting in both performance and communication, which is not a fair comparison.

Minor comments:

1.	In Page 2, first paragraph, Localized SAM is introduced first and then “sharpness-aware minimization (SAM (Foret et al., 2021))” is repeated again. It would be better to revise it.

2.	Page 2, second paragraph in Related Work,  the Walkman algorithm (Mao et al., 2020) is solved by ADMM, with two versions, one is to solve a local optimization problem, the other is to solve a gradient approximation. Therefore, it is not accurate to say that “However, these works are all based on the simple SGD for decentralized optimization.”

3.	Section 3, first paragraph, in “It can often have faster convergence and better generalization than the SGD-based Algorithm 1, as will be demonstrated empirically in Section 4.1.” The “it” does not have a clear reference.

4.	In Section 3.1, you introduced $\boldsymbol{u}_k$, which was not defined previously and did not show up after Algorithm 3.

5.	Figure 6 seems to be reused from your previous LoL optimizer work.

**Questions:**

Please see the weakness stated above.

---

> ### Author Response · Authors · 2023-11-20
> **Responses on major comments (Part 1)**
>
> We would like to thank you for your constructive feedback. Here we answer your questions point by point:
>
> > Theorem 3.7 looks like a direct combination of theoretical results obtained by Adam without momentum and SAM.
>
> Theorem 3.7 analyzes the converegence of our proposed method Localized SAM, which is not a direct combination of Adam without momentum (Triastcyn et al., 2022) and SAM. The key difference is to use localized momentum and pre-conditioner for each client, instead of sending these auxiliary parameters to next client along with the model parameter. Using localized momentum and pre-conditioner creates additional difficulty in theoretical analysis, as the momentum and pre-conditioner are not always updated and synchronized. As such, theorem 3.7 cannot be obtained by simply combining the theoretical results from Adam without momentum (Triastcyn et al., 2022) and SAM. Please also see our responses regarding novelty for more discussions.
>
> > Furthermore, the proof in Appendix does not consider the convergence guarantee that could be achieved by random walk method. That is, the Markov chain is not considered.
>
> There might be some misunderstandings. In the proof in Appendix B, Theorem 3.7 considers Markov chain for the different cases of $t \mod K = 0$ (we need to change to next client) and $t \mod K \ne 0$ (we remain in the same client). The Markov chain influences the final convergence rate by $\lambda$, which depends on the transition matrix $P$ of Markov chain. We have include a new lemma (Lemma B.1 in Appendix B) in the revised version to clarify how the Markov chain affect the convergence rate from $\lambda$ and the transition matrix $P$ of Markov chain.
>
> > Note that the last equation in Page 13 is almost the same as the convergence result (Theorem 4.3, Triastcyn et al., 2022)) except it does not have the compression part.
>
> There is an error in the previous version that the last equation of our proof in Appendix B did not include terms related to sharpness-aware minimization and was inconsistent with the conclusion in the main text. We have corrected this error in our submission and the correct version is:
>
> $\frac{1}{KT} \sum_{t=0}^{KT-1} \| \nabla L(w_t) \|^2 \le O(\frac{L(w_0) -L^*}{\eta KT} + \frac{\eta \sigma^2}{K} + \eta (1-\frac{1}{K})(\sigma_l^2+\varsigma^2+G^2 + \rho^2) + \frac{\lambda \eta \sqrt{N}}{(1-\lambda)KT}).$
>
> Our conclusion contains $\rho^2$ term (step size of the perturbation step) that is related to sharpness-aware minimization. This is different from Theorem 4.3 in (Triastcyn et al., 2022) which does not include SAM in their optimization steps.
>
> > The proof also follows exactly as Triastcyn et al. (2022).
>
> Our proof for the proposed Localized SAM is different from Triastcyn et al. (2022) in the following important ways:
> - Our proof assumes that each client updates their own pre-conditioner $v^i$ , while the proof in Triastcyn et al. (2022) assumes that the pre-conditioner is synchornized (optionally with compression) across all clients.
> -  Our proof needs to include the perturbed model throughout the proof in Appendix B, while the proof in Triastcyn et al. (2022) does not consider such perturbations.
>
> > The perturbed model is not used, means that sharp awareness minimization is not analyzed, which makes me question the soundness of Theorem 3.7.
>
> There might be some misunderstandings. Indeed, the perturbed model is used throughout the proof in Appendix B. The use of perturbed model is reflected by the term $E[g_t] − \nabla L(w_t)$, which will be zero when the perturbed model in SAM is not used (as $g_t$ will be an unbiased estimator of $\nabla L(w_t)$). When using the perturbed model, it leads to an additional $\rho^2$ term that depends on SAM.
>
> > Since SAM is integrated to prevent potential overfitting, the experiment should present this effect compared with its counterpart that does not have the perturbed model. The lack of this experiment comparison would question the necessity of incorporating SAM in the proposed Localized framework.
>
> There might be some misunderstandings. Indeed, we have conducted ablation study on the choice of different hyper-parameter $\rho$’s in Figure 8, Section 4.3. Setting $\rho = 0$ reduces to the case where we do not have perturbation, and its performance is worse than more appropriate settings (e.g., $\rho = 0.05$).

---

> ### Author Response · Authors · 2023-11-20
> **Responses on major comments (Part 2)**
>
> > The simulation only shows the loss performance of the proposed algorithms and the benchmarks, however, in
> practical, we would be more interested to see the classification accuracy.
>
> We have added these results in Appendix A.1, as are also shown here. These results are generally consistent with the loss performances (Figure 2-5 in the main text): SGD and Walkman generally achieve slightly worse final performances than other methods. Among all methods that are based on adaptive optimizers, Localized SAM achieves the best overall performance.
>
> | Graph type         |      | Ring |      |       |3-regular |     |
> |--------------------|------|------|------|------|-------|------|
> | # clients          | 10   | 20   | 50   | 10   | 20        | 50   |
> | SGD                | 82.3 | 81.7 | 81.4 | 83.2 | 82.5      | 81.9 |
> | Adam               | 85.2 | 85.0 | 84.6 |  88.8    | 88.1 | 87.4   |
> | Walkman            | 81.6 | 78.2 | 78.9 | 85.7 | 85.3 | 84.8  |
> | Adam (no momentum) |   85.1 | 84.8 | 85.2 | 88.2 | 87.7 | 87.2   |
> | Gossip             |  85.6 | 85.3 | 84.9 | 88.5 | 88.2 | 87.8  |
> | FedAvg             |  85.4 | 85.5 | 85.0 | 88.3 | **88.6** | 87.6  |
> | Localized SAM      |**85.8**   |  **86.0**  |  **85.4**    |  **89.0**    |  **88.6**    |**88.0**  |
>
> > The proposed algorithm is compared with FedAvg, however, for FedAvg, not all agents are communicating all
> the time, which does not make sense in the setting that FedAvg does not need to consider communication.
>
> There might be some misunderstandings. In our experiments, note that only the gossip method require all agents to communicate in
> each iteration. FedAvg only requires some of the agents to communicate with the central server in each iteration, and all
> the other methods (which are all from random-walk decentralized optimization) only requires one agent to communicate
> in each iteration. The communication cost of FedAvg is analyzed in experiments (Figure 2-5). Its communication cost is
> relatively small compared to the gossip method or Adam optimizer in random-walk setting, but still higher than Adam (no
> momentum) (Triastcyn et al. 2022) and our method Localized SAM.
>
> > That means, I suppose that if all agents in FedAvg communicate all the time, the performance of FedAvg might
> be much better than all the other methods, since there exists a coordinator, although the communication cost would
> be very high. The figures presented, however, show that Localized SAM is always better than FedAvg in the random
> sample setting in both performance and communication, which is not a fair comparison.
>
> For FedAvg, we follow the
> original setting of FedAvg and only uses some of the agents in each iteration, and the communication cost is computed
> based on the number of agents. For the proposed Localized SAM (and similarly other methods based on random-walk
> decentralized optimization), it only activates one client in each iteration, which updates the model using its local training
> data and sends the updated model to the next agent. Both methods do not require all agents to communicate in each iteration,
> and the comparison is fair across different methods. Moreover, in Figure 2-5, the performance of FedAvg are similar to
> Localized SAM with respect to the number of weight updates, but has a slightly larger communication cost.

---

> ### Author Response · Authors · 2023-11-20
> **Responses on minor comments**
>
> > In Page 2, first paragraph, Localized SAM is introduced first and then “sharpness-aware minimization (SAM
> (Foret et al., 2021))” is repeated again. It would be better to revise it.
>
> Thank you for raising this question. We have
> revised this part and removed undefined acronyms.
>
> > Page 2, second paragraph in Related Work, the Walkman algorithm (Mao et al., 2020) is solved by ADMM, with
> two versions, one is to solve a local optimization problem, the other is to solve a gradient approximation. Therefore,
> it is not accurate to say that “However, these works are all based on the simple SGD for decentralized optimization.”
>
> Thank you for raising this question. We have changed this sentence to “However, these works do not consider adaptive
> learning rates or momentum in decentralized stochastic optimization.”
>
> > Section 3, first paragraph, in “It can often have faster convergence and better generalization than the SGD-based
> Algorithm 1, as will be demonstrated empirically in Section 4.1.” The “it” does not have a clear reference.
>
> Thank you
> for mentioning this. We have replaced “it” with a more direct reference “Algorithm 2”.
>
> > In Section 3.1, you introduced uk , which was not defined previously and did not show up after Algorithm 3.
>
> Thank you for mentioning this. We have revised this error in the updated version.
>
> > Figure 6 seems to be reused from your previous LoL optimizer work.
>
> This figure is different from any previous
> published works to the best of our knowledge.

---

> > ### Comment · Reviewer_w8Ss · 2023-11-22
> >
> > The reviewer thanks the authors for providing feedback, regarding to
> > - Simulation results that include accuracy comparison
> > - Explanation on the effect of $\rho$
> > - Revision of Theorem 3.7
> >
> > The reviewer still has some additional questions that hopes the authors could clarify/provide:
> > 1. "Figure 6 seems to be reused from $\textbf{your}$ previous LoL optimizer work." does not mean that your figure is the same as "previous published works". Figure 6(a) and (c) are the same as Figure 5 in the paper that you previously submitted to NeurIPS 2023, except that the x-axis has a different scale. This raises two questions:
> >
> > 1.1 The Local Lookahead (LoL) Optimizer does not have SAM, but why is the loss trend the same as the Localized SAM Optimizer. This is okay if is because you set $\rho=0$ but I did not see this description in your experiment.
> >
> > 1.2 How is the "Number of weight updates" obtained? My understanding is that, in a ring network with 20 nodes, suppose the neural network has weight of $d$, since the Localized SAM allows one node communicates every K iterations, the number of weights would be $d/K$. For Fedavg, according to your setting, every K iterations, 4 nodes would be selected to communicate, therefore the number of weights would be $8d/K$. In that way, the loss vs number of weights figures and the loss vs communication cost figures actually have the similar role to show that your proposed algorithm is more communication-efficient. Again, for Figure 5 of LoL and Figure 6 in this paper, epoch 40 corresponds to weights 8000, is that true for your experiment setting?
> > 2. Why do you sample 4 agents for communicate for FedAvg rather than letting all agents to communicate? What will be accuracy if you let all agents to communicate? Would it be better than Localized SAM if we do not consider communication efficiency? This is actually my major comment 4.
> >
> > Minor comments:
> > 1. In your proof of Theorem 3.7, you miss use $D$ and $N$ for weight dimension and number of nodes. In your definition, they should be $d$ and $n$, respectively.
> > 2. The y-label of Figure 6(a) and 6(c) should be training loss instead of testing loss.

---

> ### Author Response · Authors · 2023-11-23
> **Further responses**
>
> We would like to thank the reviewer for acknowledging our feedback on simulation results, explanation on the effect of $\rho$ as well as revision of Theorem 3.7. For your additional questions, we reply to them point by point:
>
> > "Figure 6 seems to be reused from your previous LoL optimizer work.” does not mean that your figure is the
> same as “previous published works”. Figure 6(a) and (c) are the same as Figure 5 in the paper that you previously
> submitted to NeurIPS 2023, except that the x-axis has a different scale.
>
> Since submissions rejected by NeurIPS 2023 are not made public, we could not find the Local Lookahead (LoL) submission, and it seems impossible for us to compare one figure there with our figures.
>
> > The Local Lookahead (LoL) Optimizer does not have SAM, but why is the loss trend the same as the Localized
> SAM Optimizer. This is okay if is because you set $\rho=0.0$ but I did not see this description in your experiment.
>
> As mentioned in the 4th paragraph of section 4, $\rho$ is set to 0.05 for our proposed Localized SAM optimizer unless otherwise specified (this is also highlighted in blue in our updated submission). Regarding your question on the loss trend when setting $\rho = 0.0$, we suppose the loss trends with the number of clients should indeed be similar with different values of $\rho$. As is shown in theorem 3.7, more clients (i.e., a larger $n$) always leads to slower convergence, which holds for any values of $\rho$.
>
> > How is the “Number of weight updates” obtained? My understanding is that, in a ring network with 20 nodes, suppose the neural network has weight of $d$, since the Localized SAM allows one node communicates $d$ every $K$ iterations, the number of weights would be $\frac{d}{K}$. For Fedavg, according to your setting, every $K$ iterations, 4 nodes would be selected to communicate, therefore the number of weights would be $\frac{8d}{K}$.
>
> There might be some misunderstanding. As mentioned in the last paragraph (highlighted in blue) of section 4 of this revised version, “number of weight updates” means how many times each client has updated its model parameters, not the number of model parameters.
>
> The reviewer might be thinking that for localized SAM, we send $d$ parameters to the next client after performing $K$ times of $d$ weight updates, and so the per-iteration communication cost is $\frac{d}{K}$. Similarly, for FedAvg, the reviewer might be thinking that each of the 4 clients needs to communicate twice with the central server (receiving and sending model parameters) and performs $K$ times of weight updates, and so the per-iteration communication cost is $\frac{2 \times 4 d}{K} = \frac{8d}{K}$. However, indeed, in each iteration of the proposed algorithm, one client is active and it performs $K$ local updates, and then sends its (updated) model parameter to the next client (at the end of this iteration). Take the proposed method Localized SAM (Algorithm 3) as an example, the number of iterations is $T$ and the total number of weight updates is $KT$, and it only performs $T$ rounds of communication. Hence, the interpretations of "iteration" and "weight updates" are different from that of the reviewer’s.
>
> In our experiments, since all baseline methods use the same model, we consider the *relative* communication cost, which
> counts each sending of the model parameter (i.e., the whole model) as 1. Hence, as shown in Table 1 in our submission, the relative per-iteration communication cost of Localized SAM (resp. FedAvg) is $\frac{d}{K}$(resp. $\frac{8d}{K}$).
>
> > In that way, the loss vs number of weights figures and the loss vs communication cost figures actually have
> the similar role to show that your proposed algorithm is more communication-efficient.
>
> As explained in the response above, there is some misunderstanding. First, note that the figures show the loss vs the number of **weight updates**, not with the number of **weights**. Also, as explained above, each client performs $K$ weights updates before one round of communication. Hence the number of weight updates is very different from the communication cost, and that explains why the two sets of figures are so different.
>
> > Again, for Figure 5 of LoL and Figure 6 in this paper, epoch 40 corresponds to weights 8000, is that true
> for your experiment setting?
>
> Since submissions rejected by NeurIPS 2023 are not made public, we could not comment on the figure there. Also, we do not use "epoch" in this submission, as its definition may be ambiguous in decentralized optimization, and it is also not used in previous works (such as (Sun et al. 2022; Triastcyn et al. 2022; Mao et al. 2020; McMahan et al. 2017)).

---

> > ### Author Response · Authors · 2023-11-23
> > **Further responses (cont.)**
> >
> > > Why do you sample 4 agents for communicate for FedAvg rather than letting all agents to communicate?
> >
> > FedAvg (McMahan et al. 2017) and later works on federated optimization (Reddi et al. 2021; Sun et al. 2023) only sample
> > some agents (instead of using all clients) in each iteration. In our experiments, we choose 4, which is the same as in the
> > implementation of FedAvg from https://github.com/orion-orion/FedAO.
> >
> > References:
> >
> > Reddi et al. Adaptive Federated Optimization. ICLR 2021.
> >
> > Sun et al. Dynamic Regularized Sharpness Aware Minimization in Federated Learning: Approaching Global Consistency
> > and Smooth Landscape. ICML 2023.
> >
> > > What will be accuracy if you let all agents to communicate? Would it be better than Localized SAM if we do not consider communication efficiency? This is actually my major comment 4.
> >
> > As suggested, we add an experiment on FedAvg using all clients (the number of communicating clients in each iteration is 20). We compare its performance and communication cost with Localized SAM on 3-regular graph with 20 clients and original FedAvg (that samples 4 clients). The final testing accuracies as well as their relative per-iteration communication costs are shown in the table below. We can see that letting all agents to communicate only leads to marginal improvements, while the communication cost becomes significantly larger.
> >
> > | Method         | Accuracy | relative per-iteration communication cost |
> > |----------------|----------|-------------------------------------------|
> > | FedAvg (4 clients) | 88.6     |                   $\frac{2 \times 4}{5} = 1.6$                        |
> > | FedAvg (all clients)  | **88.8**     |                $\frac{2 \times 20}{5} = 8$                            |
> > | Localized SAM  | 88.6     |                   $\frac{1}{5} = 0.2$                        |
> >
> > > Minor comments
> >
> > Thank you for noting these errors. We have revised them in the updated version. We are also happy to address any further questions, and we will be more than grateful if you could kindly consider raising your score.

---

### Official Review · Reviewer_zbhE · 2023-11-02

**Soundness:** 3 good
**Presentation:** 3 good
**Contribution:** 2 fair
**Rating:** 6
**Confidence:** 3

**Summary:**

This paper proposes a communication-efficient random walk optimization algorithm for decentralized learning. The key ideas are: 1) Eliminate communication of auxiliary parameters like momentum and preconditioner in adaptive optimizers like Adam; 2) Perform multiple local model updates before communication to reduce communication frequency; 3) Incorporate sharpness-aware minimization to avoid overfitting during multiple local updates. Theoretically, it is shown that the proposed method achieves the same convergence rate as existing methods but with lower communication cost. Experiments on CIFAR-10 and CIFAR-100 demonstrate that the proposed method achieves comparable performance to Adam-based methods but with much lower communication cost.

**Strengths:**

* The method effectively reduces communication cost in decentralized learning without hurting convergence, which is important for bandwidth-limited decentralized applications.
* Convergence analysis follows standard assumptions and provides insights into how the hyperparameters affect convergence.
* Comprehensive experiments compare with reasonable baselines, evaluate different network structures and sizes, and study sensitivity to key hyperparameters.
* The paper is clearly motivated, easy to follow and technically sound overall. Figures are informative.

**Weaknesses:**

* Each component (e.g. local updates, SAM) has been studied before in different contexts. The novelty lies more in the combination.
* Missing References
  * "Can decentralized algorithms outperform centralized algorithms? a case study for decentralized parallel stochastic gradient descent"
  * "Asynchronous decentralized parallel stochastic gradient descent" where multiple local model updates is used in asynchronous decentralized training

**Questions:**

* Have you experimented with more complex network topologies beyond ring and expander graphs? There is a "chord network" topology in "Asynchronous decentralized parallel stochastic gradient descent" that may be beneficial.
* Does the algorithm work with dynamic network topology where each iteration has a different communication topology?
* Is it possible to combine your approach with compression of model parameters for further communication reduction without making the convergence rate worse?

---

> ### Author Response · Authors · 2023-11-20
> **Responses**
>
> We would like to thank you for your constructive feedback. Here we answer your questions point by point:
>
> > Each component (e.g. local updates, SAM) has been studied before in different contexts. The novelty lies more in
> the combination.
>
> While we agree that local updates or SAM has been studied before in different contexts, we would like
> to emphasize that a key component of our method is to use localized momentum and pre-conditioner, which has not been
> explored in previous works (Sun et al. 2022; Triastcyn et al. 2022) on random-walk decentralized optimization. Please also
> see our responses regarding novelty where we have listed our contributions in points.
>
> > Missing References
>
> Thank you for mentioning these works. We have added these related works into our submission.
>
> > Have you experimented with more complex network topologies beyond ring and expander graphs? There is
> a ”chord network” topology in ”Asynchronous decentralized parallel stochastic gradient descent” that may be
> beneficial.
>
> Thank you for your suggestion. As suggested, we have added experiments on the chord network with 20 clients.
> Results are shown in Figure 10 in Appendix A.2, and are generally similar to the ring graph. In general, methods based
> on adaptive optimizers outperform SGD or Walkman, regardless of the type of communication (centralized, random-walk,
> or gossip). The gossip method has a much higher communication cost than both centralized and random-walk methods.
> Walkman achieves slightly better performance than SGD with the same communication cost, but its performance is worse
> than that of adaptive optimizers, especially for the more difficult CIFAR-100 data set. Directly using Adam in random-walk
> optimization leads to higher communication cost than the other methods, and using Adam without momentum reduces
> communication cost while maintaining good performance. The proposed Localized SAM optimizer achieves similar
> convergence rate as other methods based on adaptive optimizers, while its communication cost is significantly smaller than
> all other methods.
>
> > Does the algorithm work with dynamic network topology where each iteration has a different communication
> topology?
>
> Thank you for raising this interesting question. Our algorithm can be directly combined with dynamic network
> topology, as it only needs local neighborhood information in each iteration. Theoretical analysis may require a little more
> work as we may need to assume the communication topology does not drastically change. We have also added some
> discussions on this extension in Appendix C.
>
> > Is it possible to combine your approach with compression of model parameters for further communication
> reduction without making the convergence rate worse?
>
> Thank you for this question. It should be straight-forward to
> combine our method with model compression for further communication reduction, and the convergence rate should also
> match methods that also employ model/gradient compression (Lin et al. 2018; Vogels et al. 2019). We have also added
> some discussions on this extension in Appendix C.
>
> References:
>
> Lin et al. Deep Gradient Compression: Reducing the Communication Bandwidth for Distributed Training. ICLR 2018.
>
> Vogels et al. PowerSGD: Practical low-rank gradient compression for distributed optimization. NeurIPS 2019.

---

### Official Review · Reviewer_bmmj · 2023-11-02

**Soundness:** 1 poor
**Presentation:** 2 fair
**Contribution:** 1 poor
**Rating:** 3
**Confidence:** 4

**Summary:**

This manuscript aims to improve the random-walk decentralized optimization by incorporating the idea of local SGD, keeping local optimizer states, and adding SAM optimization. Empirical results justify the effectiveness of the proposal.

**Strengths:**

* In general this manuscript is well-structured and the problem is well-motivated.
* Empirical results on various communication topologies with multiple decentralized optimizers are provided, in terms of computation cost and (relative) communication cost. Some ablation studies are also provided.

**Weaknesses:**

1. Limited novelty. The idea of using local SGD with local optimizer states is quite standard in the field of federated learning and distributed training. The same comment could be applied to include SAM for FL. See e.g., [1, 2] and the related work section of [1].
2. Limited theoretical contribution. The convergence analysis only considers the single worker case, and it cannot reflect the convergence of localized SAM for arbitrary decentralized communication topologies. Besides, the statement of theorem 3.7 should be more formal.
3. Limited experimental evaluations.
    * Some advanced decentralized communication topologies were not considered for the evaluation, e.g., the exponential graph [3, 4, 5]. Besides, the considered relative communication cost could be questionable, as it counts each sending of the model parameter as 1, instead of taking the network bandwidth / wallclock-time into account.
    * The hyper-parameter choice should be justified.
    * A carefully controlled evaluation should be provided. The considered baseline methods cannot guarantee a fair evaluation, as the community has a line of research on improving communication efficiency. It looks strange to directly compare an improved random walk decentralized optimizer with the other standard-form distributed optimizer.

### Reference
1. Improving the Model Consistency of Decentralized Federated Learning, ICML 2023. http://arxiv.org/abs/2302.04083
2. Improving generalization in federated learning by seeking flat minima, ECCV 2022. https://arxiv.org/abs/2203.11834
3. Exponential Graph is Provably Efficient for Decentralized Deep Training, NeurIPS 2021. https://arxiv.org/abs/2110.13363
4. Beyond Exponential Graph: Communication-Efficient Topologies for Decentralized Learning via Finite-time Convergence, NeurIPS 2023. https://arxiv.org/abs/2305.11420
5. Stochastic Gradient Push for Distributed Deep Learning, ICML 2019. https://arxiv.org/abs/1811.10792

**Questions:**

NA

---

> ### Author Response · Authors · 2023-11-20
> **Responses on novelty and theoretical contribution**
>
> We would like to thank you for your constructive feedback. Here we answer your questions point by point:
>
> ### Limited novelty.
>
> > The idea of using local SGD with local optimizer states is quite standard in the field of
> federated learning and distributed training.
>
> We agree that using local SGD with local optimizer states have been used
> in federated learning and distributed training (McMahan et al. 2017; Reddi et al. 2021). Indeed, we also mentioned this
> in Section 3.1. However, note that they all require a central server to aggregate updates from local optimizers, and the
> optimizer states are stored in the central server, not clients. On the contrary, our focus is on extending local SGD and local
> optimizer to random-walk decentralized optimization, which does not need any central server, and this has not been explored
> in the random-walk decentralized optimization literature (such as (Sun et al. 2022; Triastcyn et al. 2022)). In this paper, we
> demonstrate that the communication of optimizer states can be removed without performance degradation.
>
> Reference: Reddi et al. Adaptive federated optimization. ICLR 2021.
>
> > The same comment could be applied to include SAM for FL. See e.g., [1, 2] and the related work section of [1].:
>
> Although they all use SAM in distributed learning, these works focus on different types of optimization algorithms. [1]
> is based on gossip decentralized algorithms, where all clients must always be active during the training process, and [2]
> considers centralized federated learning, i.e., a central server can coordinate the whole learning process. Moreover, most of
> related works discussed in [1] either focus on centralized federated learning (Qu et al., 2022; Sun et al., 2023) or gossip
> decentralized algorithms (Sun et al. 2022; Dai et al. 2022), which are all different from our method. Our method focuses on
> random-walk decentralized optimization that does not need a central server and only require one client to be active in each
> iteration. As such, it has much weaker requirements on the communication network. Please also see our general responses
> on novelty part.
>
> References:
>
> Qu et al. Generalized federated learning via sharpness aware minimization. ICML 2022.
>
> Sun et al. Decentralized federated averaging. IEEE TPAMI 2022.
>
> Dai et al. DisPFL: Towards communication-efficient personalized federated learning via decentralized sparse training.
> ICML, 2022.
>
> Sun et al. Dynamic Regularized Sharpness Aware Minimization in Federated Learning: Approaching Global Consistency
> and Smooth Landscape. ICML 2023.
>
> ### Limited theoretical contribution.
>
> > The convergence analysis only considers the single worker case
>
> There might be some misunderstandings. Note that in random-walk decentralized optimization algorithms (Mao et al. 2020; Sun et al.
> 2022; Triastcyn et al. 2022), only one single worker is activated and updated in each iteration. Hence, we only consider one
> single worker in the convergence analysis.
>
> > and it cannot reflect the convergence of localized SAM for arbitrary decentralized communication topologies.
>
> There might be some misunderstanding, possibly because some notations have been abused in our submission. Note that in
> Section 3.2, we used $\lambda$ to denote the compression ratio in (Triastcyn et al. 2022). In theorem 3.7, we used $\lambda$ again, but to
> denote a particular eigenvalue (detailed definition can be found below) of the network’s probability transition matrix $P$. Hence, the convergence rate in theorem 3.7
> does depend on the network via $\lambda$. We corrected this and revised the definition of $\lambda$ in theorem 3.7.
>
> Definition of $\lambda$: Denote the eigenvalues of the transition matrix $P$ as
> $\lbrace \lambda_i \rbrace_{i=1}^n$,
> which satisfy $1 = \lambda_1 > \lambda_2 > \dots > \lambda_n > -1$.
> $\lambda$ is defined as $\lambda = \max(\lambda_2, |\lambda_n|)$.
>
> > Besides, the statement of theorem 3.7 should be more formal.
>
> As suggested, we have revised the statement of Theorem 3.7
> to include more detailed explanations on notations and pre-requisite of this
> theorem.
> The revised version is as follows:
>
> **Theorem 1.** Under Assumption 3.1 to 3.6, $w_t$'s generated from Algorithm 3 satisfy that:
>
> $\frac{1}{KT} \sum_{t=0}^{KT-1} || \nabla L(w_t) ||^2  \le O(\frac{L(w_0) - L^*}{\eta KT} + \frac{\eta \sigma^2}{K} + \eta
> (1-\frac{1}{K}) (\sigma_l^2+\varsigma^2+G^2+\rho^2) +  \frac{\lambda \eta \sqrt{n}}{(1-\lambda)KT})$
>
> where we denote the eigenvalues $\lbrace \lambda_i \rbrace_{i=1}^n$ of transition matrix $P$ as $1 = \lambda_1 > \lambda_2 > \dots > \lambda_n > -1$ and $\lambda = \max(\lambda_2, |\lambda_n|)$.

---

> ### Author Response · Authors · 2023-11-20
> **Responses on experimental evaluations**
>
> > Some advanced decentralized communication topologies were not considered for the evaluation, e.g., the
> exponential graph [3, 4, 5].
>
> Thank you for your comment. As suggested, we have added experiments on exponential
> graphs with 20 clients. Results are shown in Figure 9 in the revised Appendix A.2, and are generally similar to ring or
> 3-regular expander graphs. In general, methods based on adaptive optimizers outperform SGD or Walkman, regardless of
> the type of communication (centralized, random-walk, or gossip). The gossip method has a much higher communication
> cost than both centralized and random-walk methods. Walkman achieves slightly better performance than SGD with the
> same communication cost, but its performance is worse than that of adaptive optimizers, especially for the more difficult
> CIFAR-100 data set. Directly using Adam in random-walk optimization leads to higher communication cost than the other
> methods, and using Adam without momentum reduces communication cost while maintaining good performance. The
> proposed Localized SAM optimizer achieves similar convergence rate as other methods based on adaptive optimizers, while
> its communication cost is significantly smaller than all other methods.
>
> > Besides, the considered relative communication cost could be questionable, as it counts each sending of the
> model parameter as 1, instead of taking the network bandwidth / wallclock-time into account.
>
> Counting the total
> amount of transmitted data are common practice for measuring communication cost in literature (Sun et al. 2022; Triastcyn
> et al. 2022; Wang et al. 2022; Wu et al. 2022; Gao et al. 2021). As for the network bandwidth or wallclock time, note that
> the wallclock time depends on the network bandwith as well as the amount of data sent in each iteration. The network
> bandwidth should be assumed same across different algorithms for fair comparison, hence the wallclock time solely depends
> on the amount of data sent in each iteration, as is measured by the relative communication cost.
>
> References:
>
> Wang et al. ProgFed: Effective, Communication, and Computation Efficient Federated Learning by Progressive Training.
> ICML 2022.
>
> Wu et al. SmartIdx: Reducing Communication Cost in Federated Learning by Exploiting the CNNs Structures. AAAI 2022.
>
> Gao et al. On the Convergence of Communication-Efficient Local SGD for Federated Learning. AAAI 2021
>
> > The hyper-parameter choice should be justified.
>
> The hyper-parameters in our proposed method can be divided into
> two parts: (i) those that are specific to our method ($K, \rho$), and (ii) those that also exist for other methods ($\eta, \delta$). For $K$ and $\rho$, their values are chosen by a validation set and we also included studies on their sensitivity in Section 4.3 (Figure 7 for
> $K$ and Figure 8 for $\rho$). For $\eta$ and $\delta$, their choices follow previous works on decentralized optimization (Sun et al. 2022;
> Triastcyn et al. 2022). We have also included these justifications in the revised version.
>
> > A carefully controlled evaluation should be provided. The considered baseline methods cannot guarantee a
> fair evaluation, as the community has a line of research on improving communication efficiency. It looks strange
> to directly compare an improved random walk decentralized optimizer with the other standard-form distributed
> optimizer.
>
> Other techniques that improves communication efficiency include model quantization (Alistarh et al. 2017;
> Nadiradze et al. 2021), top-K sparsification (Wang et al. 2023) and gradient compression (Lin et al. 2018; Vogels et
> al. 2019). Note that our work focuses on reducing the communication cost by removing the communication of auxiliary
> parameters in adaptive optimizers. Our experiments verify that removing these auxiliary parameters does not harm the final
> performance, while significantly reducing the communication cost. Thus, these existing techniques are orthogonal to the
> proposed method and can be straightforward combined. We have also added some discussions on potential future works in
> Appendix C that combine the proposed method with other techniques.
>
> References:
>
> Alistarh et al. QSGD: Communication-Efficient SGD via Gradient Quantization and Encoding. NIPS 2017
>
> Nadiradze et al. Asynchronous Decentralized SGD with Quantized and Local Updates. NeurIPS 2021
>
> Wang et al. CocktailSGD: Fine-tuning Foundation Models over 500Mbps Networks. ICML 2023
>
> Lin et al. Deep Gradient Compression: Reducing the Communication Bandwidth for Distributed Training. ICLR 2018
>
> Vogels et al. PowerSGD: Practical low-rank gradient compression for distributed optimization. NeurIPS 2019

---

> ### Comment · Reviewer_bmmj · 2023-11-20
> **Response to the authors' feedback**
>
> The reviewer thanks the authors for providing feedback, regarding to
> * revise the manuscript and explain the meaning of $\lambda$ in their Theorem 3.7.
>
> The reviewer still has some additional questions that he/she hopes the authors could clarify/provide:
> * The technical difficulty of adding well-known SAM and local SGD to the random-walk decentralized learning algorithms, either empirically or theoretically.
> * Measure the communication cost in terms of average outward edges in the decentralized system, rather than the total number of communication edges in the whole system, as these communications could happen simultaneously. The latter one (used in the current manuscript) is a bit unfair and could naturally bias the random-walk decentralized learning algorithm. The reviewer is curious about the comparison between localized SAM and D-SGDm on the exponential graph.
> * It is a bit unclear which exponential graph is used in Figure 9, as Takezawa et al 2023 is still different from that of Ying et al 2019. More clarification should be provided, together with the accuracy results, rather than the loss results.
> * The reviewer is also curious about one crucial ablation study: what if we just straightforwardly apply SAM to all other baselines (especially the SOTA method, namely DSDGm + exponential graph), and compare the corresponding results with Localized SAM?

---

> ### Author Response · Authors · 2023-11-22
> **Further responses**
>
> Thank you for acknowledging our responses. Here we provide responses for your further questions point by point:
>
> > The technical difficulty of adding well-known SAM and local SGD to the random-walk decentralized learning
> algorithms, either empirically or theoretically.
>
> Introducing SAM to random-walk decentralized learning algorithms
> creates additional difficulty in theoretical analysis, as the model update $g_t$ is no longer an unbiased estimator of gradient
> $\nabla L(w_t)$. As such, we need to upper-bound its difference $E[g_t] - \nabla L(w_t)$, which leads to an additional $\rho^1$ term that depends on SAM in the final results in Theorem 3.7. We would also like to emphasize that our method is not simply adding well-known SAM and local SGD to random-walk decentralized learning algorithms. A key difference is to use localized
> momentum and pre-conditioner for each client, instead of sending these auxiliary parameters to next client along with the
> model parameter. Using localized momentum and pre-conditioner creates additional difficulty in theoretical analysis as well,
> as the momentum and pre-conditioner are not always updated and synchronized. Please also see our responses regarding
> novelty as well as our responses to reviewer w8Ss for discussions on this point.
>
> > Measure the communication cost in terms of average outward edges in the decentralized system, rather than the
> total number of communication edges in the whole system, as these communications could happen simultaneously.
> The latter one (used in the current manuscript) is a bit unfair and could naturally bias the random-walk decentralized
> learning algorithm. The reviewer is curious about the comparison between localized SAM and D-SGDm on the
> exponential graph.
>
> Thank you for your suggestion. We have added such additional results for exponential graph in Appendix A.1. Under this metric, the relative per-iteration communication cost for D-SGDm on the exponential graph will be $\lfloor \log(20) \rfloor +1 = 5$, which is still higher than that of most random-walk decentralized learning algorithms shown in Table 1. And from the newly-added Figure 12, D-SGDm (referred as gossip method) still has higher communication cost than random-walk decentralized learning algorithms.
>
> > It is a bit unclear which exponential graph is used in Figure 9, as Takezawa et al 2023 is still different from that
> of Ying et al 2019. More clarification should be provided
>
> We have mentioned that we use the static exponential graph defined in (Ying et al. 2021). For easier reference, we have also plotted static exponential graph with 20 clients in the revised Figure 9(a).
>
> > together with the accuracy results, rather than the loss results.
>
> The accuracy results are added to Table 2 in Appendix A.1 in the revised version. These results are generally consistent with the loss performances (Figure 2-5 in the main text) as well as other types of graphs (ring, 3-regular or chord). SGD and Walkman generally achieve slightly worse final performances than other methods. Among all methods that are based on adaptive optimizers, Localized SAM achieves the best overall performance.
>
> > The reviewer is also curious about one crucial ablation study: what if we just straightforwardly apply SAM to all
> other baselines (especially the SOTA method, namely DSDGm + exponential graph), and compare the corresponding
> results with Localized SAM?
>
> As suggested, we conducted some new experiments to apply SAM to DSGDm on the exponential graph. This new baseline (refered to as DSGDm+SAM) is added to our newly revised Figure 10 and 12 in Appendix A.2, and its accuracy on exponential graph with 20 clients is added to Table 2 in Appendix A.1 (also shown below). We can see that DSGDm+SAM has a better performance than Gossip/DSGDm, and also matches the performance of Localized SAM. However, the communication cost of DSGDm+SAM is the same as DSGDm, and is still much higher due to its gossip-type communication, as can be seen from Figures 10 and 12 in Appendix A.2.
>
> | Method         | Accuracy |
> |----------------|----------|
> | Gossip (DSGDm) | 88.5     |
> | DSGDm+SAM      | **88.7**     |
> | Localized SAM  | **88.7**     |
>
> We are happy to address any further questions, and we will be more than grateful if you could kindly consider raising your score.

---

> > ### Comment · Reviewer_bmmj · 2023-11-22
> > **additional questions**
> >
> > The reviewer thanks the authors for providing the followup results. The reviewer has one followup question:
> > * it is possible to consider the 1-peer time-varying exponential graph discussed in Ying et al 2021 or Base-2 graph (1) introduced in Takezawa et al 2023, instead of the communication-intensive static exponential graph? this type of communication topology can significantly reduce the communication cost while ensuring a similar performance as a static exponential graph.

---

> ### Author Response · Authors · 2023-11-23
> **Reply to the additional question**
>
> Thank you for your suggestion. We use the static graph to ensure a fair comparison between random-walk and gossip-type algorithms. While using these dynamic graphs might reduce the communication cost of gossip-type algorithms, in practice the communication graph is often defined by the application scenario and remains static during model training, and so we focus on the case of static graphs for different methods. Please also see our responses to reviewer zbhE who has raised a similar question on how random-walk algorithms can be applied to dynamic graphs, and some discussions on this possible future direction can be found in Appendix C.
>
> We are happy to address any further questions, and we will be more than grateful if you could kindly consider raising your score.

---

### Author Response · Authors · 2023-11-20
**General responses on novelty**

We would like to first thank all reviewers for your detailed comments and constructive feedbacks. We note that most reviewers have concerns on novelty of our submission, and we would like to emphasize that our work focus on random-walk decentralized
optimization, which is different from centralized methods that require a central server or gossip-type decentralized optimization that requires all clients to be active. The contributions of our method are summarized at the end of Section 1 in our
submission, and we re-state some key points related to novelty here:

- Reduce communication cost: we are the first to consider removing the communication of auxiliary parameters to reduce
the communication cost in random-walk decentralized optimization
- SAM in random-walk decentralized optimization: we are the first to introduce sharpness-aware minimization to random-
walk decentralized optimization, which alleviates potential over-fitting when performing multiple local updates on the
same client.
- Theoretical analysis: we show theoretically that the proposed method can converge faster than existing works under the
same communication cost.

---

### Meta-Review · Area_Chair_tUVU · 2023-12-10

**Metareview:**

The paper proposes a random-walk Adam-based optimizer for decentralized learning, aiming at improving communication efficiency.
Unfortunately consensus among the reviewers remained that it remains slightly below the bar after the discussion phase. Main remaining concerns were on the level of technical novelty provided of the proofs as a close consequence of known results for the respective building blocks (such as Adam without momentum or SAM), as well as scaling with many workers. Also, questions remained on the setup of baselines (e.g. FedAvg) in terms of communication frequency.

We hope the detailed feedback helps to strengthen the paper for a future occasion.

**Justification For Why Not Higher Score:**

Not enough for the high bar of ICLR, given that the level of novelty on the theoretical side wasn't strong enough at this point

**Justification For Why Not Lower Score:**

N/A

---

### Decision · Program_Chairs · 2024-01-16

Reject